# Crossroads between Skin and Endocrine Glands: The Interplay of Lichen Planus with Thyroid Anomalies

**DOI:** 10.3390/biomedicines12010077

**Published:** 2023-12-28

**Authors:** Andreea-Maria Radu, Mara Carsote, Claudiu Nistor, Mihai Cristian Dumitrascu, Florica Sandru

**Affiliations:** 1Department of Dermatovenerology, Elias University Emergency Hospital, 011461 Bucharest, Romania; 2Department of Endocrinology, Carol Davila University of Medicine and Pharmacy, 050474 Bucharest, Romania; 3Department of Clinical Endocrinology V, C.I. Parhon National Institute of Endocrinology, 020021 Bucharest, Romania; 4Department 4—Cardio-Thoracic Pathology, Thoracic Surgery II Discipline, Carol Davila University of Medicine and Pharmacy, 050474 Bucharest, Romania; 5Thoracic Surgery Department, Dr. Carol Davila Central Military Emergency University Hospital, 010242 Bucharest, Romania; 6Department of Obstetrics and Gynaecology, C. Davila University of Medicine and Pharmacy & University Emergency Hospital, 050474 Bucharest, Romania; drdumitrascu@yahoo.com; 7Department of Dermatovenerology, Carol Davila University of Medicine and Pharmacy & Elias University Emergency Hospital, 011461 Bucharest, Romania; florysandru@yahoo.com

**Keywords:** lichen planus, thyroid, thyroiditis, hypothyroidism, thyroid nodule, thyroid cancer, thyroidectomy, antibody, skin, oral lichen planus

## Abstract

In this narrative review, we aimed to overview the interplay between lichen planus (LP) and thyroid conditions (TCs) from a dual perspective (dermatologic and endocrine), since a current gap in understanding LP-TC connections is found so far and the topic is still a matter of debate. We searched PubMed from Inception to October 2023 by using the key terms “lichen planus” and “thyroid”, (alternatively, “endocrine” or “hormone”). We included original clinical studies in humans according to three sections: LP and TC in terms of dysfunction, autoimmunity, and neoplasia. Six studies confirmed an association between the thyroid dysfunction (exclusively hypothyroidism) and LP/OL (oral LP); of note, only one study addressed cutaneous LP. The sample size of LP/OLP groups varied from 12–14 to 1500 individuals. Hypothyroidism prevalence in OLP was of 30–50%. A higher rate of levothyroxine replacement was identified among OLP patients, at 10% versus 2.5% in controls. The highest OR (odd ratio) of treated hypothyroidism amid OLP was of 2.99 (*p* < 0.005). Hypothyroidism was confirmed to be associated with a milder OLP phenotype in two studies. A single cohort revealed a similar prevalence of hypothyroidism in LP versus non-LP. Non-confirmatory studies (only on OLP, not cutaneous LP) included five cohorts: a similar prevalence of hypothyroidism among OLP versus controls, and a single cohort showed that the subjects with OLP actually had a lower prevalence of hypothyroidism versus controls (1% versus 4%). Positive autoimmunity in LP/OLP was confirmed in eight studies; the size of the cohorts varied, for instance, with 619 persons with LP and with 76, 92, 105, 108, 192, 247, and 585 patients (a total of 1405) with OLP, respectively; notably, the largest control group was of 10,441 individuals. Four clusters of approaches with respect to the autoimmunity in LP/OLP were found: an analysis of HT/ATD (Hashimoto’s thyroiditis/autoimmune thyroid diseases) prevalence; considerations over the specific antibody levels; sex-related features since females are more prone to autoimmunity; and associations (if any) with the clinical aspects of LP/OLP. HT prevalence in OLP versus controls was statistically significantly higher, as follows: 19% versus 5%; 12% versus 6%; and 20% versus 9.8%. A single study addressing LP found a 12% rate of ATDs. One study did not confirm a correlation between OLP-associated clinical elements (and OLP severity) and antibody values against the thyroid, and another showed that positive TPOAb (anti-thyroperoxidase antibodies) was more often found in erosive than non-erosive OLP (68% versus 33%). Just the reverse, one cohort found that OLP subjects had a statistically significantly lower rate of positive TPOAb versus controls (9% versus 15%). Five case-control studies addressed the issue of levothyroxine replacement for prior hypothyroidism in patients that were diagnosed with OLP (no study on LP was identified); three of them confirmed a higher rate of this treatment in OLP (at 8.9%, 9.7%, and 10.6%) versus controls. In conclusion, with regard to LP/OLP-TC, we note several main aspects as practical points for multidisciplinary practitioners: OLP rather than LP requires thyroid awareness; when it comes to the type of thyroid dysfunction, mostly, hypothyroidism should be expected; female patients are more prone to be associated with ATDs; a potential higher ratio of OLP subjects taking levothyroxine was found, thus a good collaboration with an endocrinology team is mandatory; and so far, OLP individuals have not been confirmed to be associated with a higher risk of thyroid nodules/cancer.

## 1. Introduction

Lichen planus (LP), a chronic inflammatory condition usually affecting middle-aged individuals with an overall prevalence of 0.14 to 1.27%, associates three main clinical subtypes: cutaneous, mucosal, and planopilaris (located at the level of the scalp) [1,2,3]. Cutaneous LP involves 0.4% to 1.2% of all referrals in the field of dermatology [4,5]. Clinically, this form appears as an eruption characterised by the presence of flat-topped, violaceous, papular lesions of different sizes that are generally described as the “six P’s” (standing for “purple colour, pruritic nature, polygonal shape, planar appearance, papules, and plaques”). Additionally, Wickham striae have been described [6,7,8]. The eruption’s distribution tends to be confined to the extremities; however, there are some instances when it may be generalized [8,9,10]. Clinical remission is registered within one to two years in the most LP cases [4,11,12].

Oral LP (OLP) serves as the mucosal counterpart to LP, despite the notable clinical diversity seen between the two. While skin lesions are often self-limited, OLP is characterised by a chronic nature, infrequent spontaneous remission, and a higher risk of pre-malignancy/malignancy, and increased morbidity, still being a challenge in terms of health care [12,13,14]. Isolated occurrences of OLP are often seen in dentistry settings, while 20% of OLP patients also exhibit synchronous or asynchronous cutaneous lesions [13,15]. The six clinical subtypes of OLP (reticular, atrophic, plaque-like, papular, erosive/ulcerative, and bullous) may manifest either in isolation or in association with another type; the reticular, erosive/ulcerative, and plaque-like variants are the most frequent variants. Reticular lesions, widely acknowledged as the predominant manifestation of OLP, often exhibit no symptoms but cause many papules with a network of tiny, elevated, whitish-grey, lace-like lesions (Wickham striae). Atrophic and erosive/ulcerative OLP plaques may induce local discomfort, while a plaque subtype displays leukoplakia-like features (white, uniform, slightly raised, multiple, and smooth lesions, especially at the tongue and buccal mucosa) [13,16,17,18].

### 1.1. Pathogenic Traits of LP/OLP

LP/OLP has been correlated with various systemic conditions (underlying different pathogenic features that are more or less understood so far), including type 2 diabetes mellitus, arterial hypertension, metabolic syndrome, psychosomatic disorders, chronic liver and gastrointestinal diseases (including celiac disease), etc. [13,16]. Moreover, a potential correlation between LP/OLP and thyroid disorders, particularly hypothyroidism and autoimmune thyroiditis, has been reported, noting the general frame of multidisciplinary, autoimmune conditions that have a tendency to coexist due to common mechanisms [19,20,21].

While various pathogenic elements have been studied in LP/OLP (genetic and environmental, including nutrients, hormonal, immune/autoimmune, molecular, and biochemical ones such as oxidative stress, etc.), the exact cause of LP/OLP remains rather unknown [21,22]. A familial occurrence has been noted, as well, but a clarification of this specific genetic background is still needed [23,24,25]. For instance, we note the potential role of the initial intron within the promoter gene of interferon-gamma (IFN-γ) [26] or pro-angiogenetic factors in OLP [27], respectively, of 308A allele of tumour necrosis factor-alpha (TNF-α) in cutaneous LP [26].

Interestingly, psychological factors might contribute to the development of OLP, as elements such as anxiety or depression might enhance the risk of OLP/LP, but a dual component is actually involved: the clinical presentation of OLP/LP may represent the trigger of such elements, thus impairing the overall quality of life [28,29], or an episode of OLP exacerbation may be registered amidst periods of increased psychological stress [30,31,32]. Also, prior trauma serves as a contributor to OLP/LP onset in high-risk individuals [21].

### 1.2. The Thyroid Gland: A Player in the Field of Dermatologic Conditions

The thyroid gland has a crucial role in regulating several important physiological processes such as growth, development, tissue repair, and metabolic activities. Abnormalities of thyroid hormone levels are described in a plethora of multidisciplinary conditions, including in dermatology, whereby affected patients often experience a significant discomfort and a reduced quality of life for various reasons [33]. Overall, thyroid conditions should be regarded upon three main chapters: dysfunctionality (namely, hyperthyroidism and hypothyroidism); autoimmune thyroid diseases (ATDs) underlying either normal or abnormal functions; and thyroid nodules, including those displaying malignancy traits in less than 5% of them (usually, thyroid neoplasia is accompanied by a normal thyroid function, unless an ATD-associated dysfunction is overlapped).

ATDs, with an overall prevalence of 5%, mainly include two autoimmune conditions depending on the antibody profile: on the one hand, there is chronic autoimmune (lymphocytic) Hashimoto’s thyroiditis (HT) which carries an increased risk of hypothyroidism due to the presence of thyroid-blocking antibodies, specifically, anti-thyroperoxidase antibodies (TPOAbs) and anti-thyroglobulin antibodies (TgAbs), but euthyroidism and transitory (flare-up) thyrotoxicosis may be found as well, and, on the other hand, Basedow–Graves’ disease (GD) that is characterised by the presence of thyroid-stimulating immunoglobulins of Thyroid-Stimulating Hormone (TSH) receptor antibodies (TRAbs) causing a TSH suppression (hyperthyroidism) [34,35].

In addition to this panel of antibodies against the thyroid, numerous factors are incriminated in ATD-associated pathways such as infections (mostly, viral); micronutrients; a discontinuation of glucocorticoid therapy; stress; and genetic susceptibility, including genes associated with the human leukocyte antigen (HLA) system, etc. [36,37,38]. Due to common autoimmune mechanisms, similar extra-thyroid conditions are found in ATD individuals such as (endocrine diseases) Addison’s disease, premature ovarian failure, type 1 diabetes mellitus, autoimmune hyphophysitis, and hypoparathyroidism, respectively, and (non-endocrine disorders) celiac disease, rheumatoid arthritis, lupus, vitiligo, and autoimmune alopecia, etc. [39,40,41,42].

### 1.3. Aim

We aimed to overview the interplay between LP and different thyroid conditions from a dual clinical perspective (dermatology and endocrinology approaches), having, as starting point, either the adult population diagnosed with LP/OLP or the one confirmed with a thyroid anomaly. The motivation of this topic is based on the current gap in understanding the connection between LP/OLP and the thyroid panel. According to our current knowledge, LP/OLP does not represent a single disease with a clear pathogenic component but is rather a type of immune response to a multitude of environmental elements (which are more or less understood so far) in genetically susceptible subjects. Whether thyroid anomalies are a part of these LP/OLP contributors and triggers or whether they are also a consequence of the same environmental factors is less clear. Apart from pathogenic loops, our hypothesis involves clinical aspects, namely, the fact that some population subgroups diagnosed with LP/OLP are at a higher risk of having hypothyroidism as well as positive thyroid serum antibodies than controls.

## 2. Methods

This study’s design was of a narrative review. We searched the PubMed database from Inception to October 2023, particularly using the key research terms “lichen planus”, “thyroid”, “endocrine”, and “hormone”. The inclusion criteria were original, clinical (non-experimental) studies in humans (clinical studies); English papers; and studies that provided the traditional thyroid panel of evaluation (any of the following: thyroid function, serum antibodies, imaging features such as ultrasounds, and pathological reports in some cases). Exclusion criteria were animal studies and interventional trials; non-English papers; and other types of articles such as case reports, editorials, letters to editor, and reviews.

We followed three main sections: LP/OLP and anomalies of the thyroid in terms of dysfunctionality (hypothyroidism and hyperthyroidism); thyroid autoimmunity (ATDs, namely HT and GD); and neoplasia (nodules and/or cancer). Particularly, the data were displayed according to the followings subsections: the association between LP/OLP and thyroid function anomalies (6 confirmatory studies and 5 non-confirmatory cohorts); the spectrum of positive thyroid autoimmunity amid LP/OLP (8 confirmatory and one non-confirmatory study); the use of levothyroxine replacement for prior hypothyroidism in patients confirmed with LP/OLP (5 studies); and the co-presence of thyroid nodules/cancer concerning the same dermatologic conditions (2 studies).

## 3. Results

### 3.1. LP/OLP: Delve into Thyroid Conditions

#### 3.1.1. LP/OLP and Thyroid Dysfunction

Regarding LP/OLP, most studies addressed the issue of hypothyroidism, rather than hyperthyroidism, while the dermatologic approach is mainly focused on OLP, not LP, in order to address the thyroid function. Overall, eleven studies addressed the issue of identifying any type of abnormal thyroid hormone levels in LP/OLP. For instance, a detailed sample-based analysis pinpointed a retrospective, case-control study conducted by Piloni et al. [43] that assessed the thyroid status in patients previously diagnosed with OLP (N1 = 76) and oral lichenoid lesions (OLLs) (N2 = 133) versus controls (N = 98) and found a higher prevalence of hypothyroidism among these two groups when compared to the control group (30.26% versus 9.18%, *p* = 0.007; 21.8% versus 9.18%, *p* = 0.0129, respectively). The findings of this cohort confirmed that hypothyroidism was statistically significantly associated with OLP; thyroid dysfunction, as a whole, has proven to be a risk factor for OLLs (oral lichenoid lesions) as well (*p* = 0.0129). Yet, the association between OLLs and hypothyroidism, as a distinct subgroup, was not statistically relevant (*p* = 0.0965). Henceforth, the correlation between OLP and hypothyroidism seemed the most important association, despite noting that a correlation does not mean causality [43].

Another cross-sectional study published in 2019 (N = 14 OLP subjects out of 860 individuals) found a statistically significantly higher prevalence of hypothyroidism among the OLP subgroup versus the entire cohort (35.7% versus 3.9%, *p* = 0.0001). Of note, patients with a co-existing mentioned thyroid anomaly and OLP exhibited a milder form of OLP, particularly characterised by a higher prevalence of reticular lesions and a lower prevalence of erythematous/ulcerative lesions in comparison to those without any thyroid illness [44]. Similarly, another cross-sectional study published 2019 by Kumar et al. [45] included 75 persons diagnosed with LP, both cutaneous and mucosal types (N = 12/75 confirmed with OLP), versus 75 controls; hypothyroidism prevalence was statistically significantly higher among the OLP subgroup (of 33.33%) in contrast to the LP subgroup (10.67%; *p* = 0.005); however, LP patients versus non-LP/OLP individuals had a similar prevalence of hypothyroidism (10.67% versus 4%, *p* = 0.117) [45].

Moreover, the identification of OLP may serve as the starting point of searching the thyroid status, and a potential identification of an abnormal TSH might not come as a surprise. For example, Roberto-Sierra et al. [46] conducted a case-control study in 2018 on 108 adults with OLP without a prior history of any thyroid condition versus 657 non-OLP individuals (healthy controls) and found a statistically significantly increased prevalence of hypothyroidism among the first group versus the controls (58.1% versus 37.3%, *p* = 0.0179), while TRAb was more prevalent in the OLP than the non-OLP subjects [46]. The same research group previously published a study enrolling OLP patients (N = 1611) and random controls from the general population (N = 1615); 11% (representing 170 subjects) of the OLP group were actually under levothyroxine supplementation, a prevalence that turned out to be 5 times higher than that seen in the controls (2.5%, meaning 40 patients who were under thyroid hormone replacement), thus confirming a higher prevalence of (treated) hypothyroidism among OLP patients (odds ratio of 2.99, 95% confidence interval between 2.03 and 4.44, *p* < 0.001). Of note, once again, the OLP subgroup with hypothyroidism had a milder form than the OLP subgroup not taking levothyroxine, meaning not having treated hypothyroidism [47], data that are consistent with the previously mentioned findings of Amato-Cuartas et al. [44] in a Colombian population [44]. Yet, when it comes to hyperthyroidism, similar results were revealed among the studied subgroups [47]. Looking at older studies, we found the one of Siponen et al. [48] that was published in 2010, showing, once again, a higher rate of hypothyroidism among OLP individuals in a Finnish population versus controls (10% versus 5%; odds ratio of 2.19, 95% confidence interval between 1.03 and 4.90) [48].

Conversely, the identification of an increased number of subjects with low thyroid hormone levels in OLP (and potentially in LP) is not unanimous. Among the ten mentioned studies that aimed to assess the thyroid status in LP/OLP individuals, we identify four of them as being non-confirmatory (each cohort confirmed with a mucosal pathology had >100 patients to a maximum of >500 individuals per article) [49,50,51,52].

Contrariwise, Tang et al. [49] showed that hypothyroidism prevalence was lower in OLP patients (N = 585) versus controls (N = 10,441), 1.2% versus 4.1% (*p* < 0.005), while hyperthyroidism had a similar rate amid the mentioned subgroups (0.5% versus 0.5%, *p* > 0.05) [49]. Kats et al. [50] identified a similar prevalence of hypothyroidism (approximately of 10%) in OLP patients (N = 102) versus controls (N = 102). The authors did not find any association between various clinical characteristics of the dermatologic condition (symptomatology, clinical type, and localization of lesions) and, respectively, hypothyroidism or any thyroid dysfunction as a matter of fact (*p* > 0.05) [50]. In 2018, a case-control study showed alike rates of both hypothyroidism and hyperthyroidism between any of the 3 subgroup OLP patients (N = 192), OLL patients (N = 122), and controls (N = 163) [51]. Another retrospective, comparative cohort conducted by Lavaee et al. [52] agreed with the findings of previous studies; 523 individuals confirmed with OLP versus 523 controls (majority being represented by females at 74% and 73.8%, respectively) were divided by the clinical type of OLP. The prevalence of hypothyroidism was the highest among individuals diagnosed with keratotic, ulcerative, atrophic, and erosive OLP (in that order). Yet, a lack of correlation between hypothyroidism and any type of OLP represented the main results. Despite the fact that individuals with OLP had a 1.7-times higher risk of developing hypothyroidism compared to the general population (odds ratio of 1.714, 95% confidence interval: 0.984–2.987), a Chi-square test did not show a statistically significant relationship between hypothyroidism and OLP [52].

As limitations of the cited studies that might bring additional biases in the interpretation of the association between thyroid dysfunction and LP/OLP, we note the following: a retrospective type of study with a prior collection of the variables [43,46,48,52]; parameters were based on a population belonging to a specific geographic area that does not allow for a generalization [43,44]; a small sample size [44]; interferences of metabolic issues, such as dyslipidaemia and metabolic syndrome, in addressing the rate of hypothyroidism and its consequences [45]; the comparison with the control group was performed according to historical (not real-time) data [49]; and the analysis of the etiological types of hypothyroidism was not performed, thus pointing to a potential pathogenic clue is not applicable [52]. Moreover, concomitant drugs might falsely supress serum TSH levels such as glucocorticoids administration and somatostatin analogues like octreotide; thus, the thyroid panel might not be accurate [50].

#### 3.1.2. LP/OLP and Positive Thyroid Autoimmunity

When facing the thyroid autoimmune background in LP/OLP, either in terms of HT or GD, a sample-based-analysis case exclusively pointed to HT rather than GD. ATDs were confirmed by the assessment of serum antibodies, and sometimes a secondary criterion of thyroid ultrasound features was added, despite not being mandatory in daily practice to sustain the diagnosis itself [42]. Among the eight confirmatory studies we identified, four of them were mentioned above, since the thyroid function was analysed as well [43,47]. An additional study we appreciated was non-confirmatory in this particular matter. The key finding remains a higher rate of positive thyroid antibodies in terms of HT diagnosis among LP/OLP individuals which sustains a certain level of awareness across dermatological practices. Secondarily, women seem more susceptible to associate with positive thyroid antibodies.

For example, a prior study of Piloni et al. [43] from 2023 provided a confirmation between the association of OLP and HT; a higher HT prevalence amid OLP (19.74%) versus controls (5.1%, *p* = 0.005), but not in OLL, was revealed [43]. The retrospective study of Anttonen et al. [53] analysed ATD rates among LP individuals, namely, a cohort of 619 subjects with a mean age of 54.2 ± 16.5 years (41.7% were males). A total of 12.4% of them had ATDs, this being statistically significantly more frequent in LP women (N = 258) compared to LP men (N = 361), 17.7% versus 5% (*p* < 0.001), and compared to the controls (8.63% and 1.56% had ATDs among the women and men, respectively) [53]. These parameters represented an additional confirmation of the traditional data showing that women are 4 to 10 times more susceptible to ATDs than males [54,55].

Similarly, Zhang’s [56] cross-sectional study from 2022 on 247 patients with OLP (average age of 45.21 ± 12.72 years) identified an HT prevalence among women (N = 186) of 46.24%, which was statistically significantly increased when compared to OLP males (N = 61, 19.67%, *p* = 0.000) [56]. The positivity rate of both TPOAbs and TgAbs was elevated in women than men (39.25% versus 19.67%, *p* = 0.005; 25.27% versus 6.56%, *p* = 0.002, respectively). No association between the clinical manifestations in OLP and HT-related antibodies was confirmed, neither when using OLP clinical scores or the description of OLP subtypes [56]. Moreover, a cross-sectional study from 2020 was conducted by Tang et al. [49] on 585 patients diagnosed with OLP from East China (mean age of 52.80 ± 13.52 years) compared to 10,441 controls (mean age of 53.50 ± 13.05 years); HT prevalence was statistically significantly higher in the studied population (12.1% versus 6.1%, *p* < 0.05) [49], as in previously mentioned cohorts [53,56]. Of similar note, Zhou et al. [51] showed, in 2018, that HT prevalence was higher in OLP adults (N = 192, 20%) versus controls (N = 163, 9.8%, *p* = 0.000) and even versus OLL patients (N = 123, 18%, *p* = 0.011) [51]. Of note, the daily diagnosis of HT may be established upon abnormally high TPOAbs and/or TgAbs for an individual (with no specific cut offs) regardless of co-morbidities [57,58].

Alikhani et al. [59] studied the severity of OLP in terms of erosive lesions with regard to the titter of serum TPOAbs; HT prevalence (defined as positive blood assays for TPOAbs for a level of more than 35 IU/mL) was increased in the subgroup with erosive (N = 44, 68%) versus non-erosive OLP (N = 48; 33%, *p* = 0.023). Similar results were pointed out with respect to interleukin-8 (IL-8) levels, thus suggesting a connection to erosive OLP, as seen in TPOAbs [59]. Notably, a prior study of Sun et al. [60] identified IL-8 as a more sensitive biomarker than IL-6 concerning OLP follow-up and an associated prognosis [60]. This cytokine is traditionally recognized in relationship with angiogenesis stimulation as well as tumour cell proliferation and migration in different malignant and non-malignant proliferative or inflammatory conditions [61,62,63].

The above cited study of Robledo-Sierra et al. [47] that was published in 2015 confirmed, one more time, a statistically significantly higher prevalence of HT among OLP patients (N = 108, 22%) when compared to controls (N = 40; 2.5%, *p* = 0.0037) [47]. Meanwhile, Lo Muzio et al. [64] identified that 14% of subjects diagnosed with OLP (N = 105) had HT (as defined by using the assays of both blocking serum antibodies and suggestive ultrasound features), which is higher than that seen in a general population (*p* < 0.0003) with a 1% prevalence of HT-related hypothyroidism, as was considered in a study from 2013 [64].

In contrast, a prior referred study from 2018 analysed four groups, OLP patients without a history of any thyroid dysfunction or levothyroxine supplementation (N = 110, mean age of 65.6 ± 9.8 years); controls (N = 657, average age of 49.4 ± 10.3 years); and two comparators, OLP subjects under levothyroxine replacement for a prior diagnosis of hypothyroidism (N = 108, mean age of 66.5 ± 10.4 years) and patients with a similar regimen of thyroxine supplementation but without OLP (N = 58, average age of 67.4 ± 14.1 years), and found no correlation between OLP and abnormal concentrations of the antibodies against the thyroid [46]. Quite the opposite with prior results [43,49,51], the prevalence of high TPOAbs was lower in OLP persons without any hormonal medication for the thyroid versus controls (9.4% versus 15%, *p* = 0.024) [46].

As limitations of these studies, we specify the following: a registry-based collection of parameters [53]; different geographic areas, such as Italy [43], Sweden [47], Finland [53], China [49,51,56], and Iran [59]; the evaluation was conducted at a hospital which is why less severe patients of LP who were followed at primary care centres were not included [53]; the inter- and intra-individual fluctuations of the serum thyroid antibodies (which usually do not serve as prognostic markers in daily endocrine practice) [49,51,57,58]. Notably, one correlation does not mean causality, while the association between LP/OLP and ATDs might be regarded amid the general panel of increased autoimmunity risk of these patients and not a specific (tide) connection between LP/OLP and ATDs [43,46,47,49,51,53,56,59,64]. Additionally, the timing of identifying positive thyroid antibodies in relationship with the confirmation of OLP should be studied, since these antibodies might activate OLP and then be decreased at the serum level; thus, a cross-sectional analysis might not be conclusive [59]. Also, a histological confirmation of HT (which is not mandatory in daily practice, neither it is commonly used) might limit the enrolment of individuals with an HT diagnosis among those already confirmed with LP/OLP [64].

#### 3.1.3. LP/OLP Patients under Levothyroxine Replacement

Remarkably, the dermatologic perspective of the potential thyroid-associated issues in LP/OLP subjects might be regarded from the angle of individuals who were treated with specific endocrine therapy, namely, thyroid hormone levothyroxine due to its deficiency (namely, hypothyroidism). However, in daily endocrine practice, this supplementation is offered for a heterogeneous panel of conditions like primary (thyroid-related) or secondary (pituitary-related) clinical/subclinical hypothyroidism, with the primary type remaining the most impactful in the general population, with a majority being caused by ATDs, such as HT, and even by the therapy for GD and/or thyroidectomy, etc. Of note, triiodothyronine usage is currently restricted to a (small) selected number of patients and it does not represent an epidemiologic issue [65,66,67].

Under these specifications, we mention that some studies distinctly analysed the status of levothyroxine therapy (as part of the endocrine management, not of the specific approach in LP/OLP) and found discordant results; yet, the pitfall of addressing this insight is represented by the non-homogenous background of underlying thyroid anomalies and by different levels of controlled/uncontrolled hypothyroidism, as well as target TSH in accordance with various guidelines, depending on the primary endocrine condition [68,69,70]. Thus, two additional case-control studies [71,72] and a prior cited one [47] showed the following in OLP individuals: Garcia-Pola et al. [71] found a statistically significant higher rate of levothyroxine usage in an OLP group (N = 215) compared to controls (N = 215), 9.7% versus 3.3% (*p* = 0.006) [71]; similar data were reported by Robledo-Sierra et al. [47] in 2015 (10.6% versus 2.6%, *p* < 0.0001) and in 2013 (8.9% versus 2.5%, *p* < 0.001) [72]. A multivariate logistic regression model pointed out an odds ratio of 3.39 for levothyroxine intake and OLP patients (95% confidence interval between 2.09 and 5.46, *p* < 0.001) [72]. Nevertheless, two other mentioned case-control studies did not confirm these results (also, exclusively in OLP, not in LP), namely, the research conducted by Kats et al. [50] in 2019 (N = 102 OLP subjects) [50] and by Siponen et al. [48] in 2010 (N = 222 OLP/OLL individuals), with a non-statistically significant odds ratio of 0.94 (95% confidence interval of 0.27–5.2) [48]. Notably, the rate of thyroid hormone supplementation in OLP patients was similar with the rate found within the first three mentioned studies [47,71,72], approximately at 10–12% [48,50].

Among the potential limitations of the cited studies, we mention the fact that survey-based data collection with respect to the levothyroxine administration might bring an additional bias [47]. Also, different geographic areas associate distinct rates of thyroxine-requiring hypothyroidism (Sweden, Finland, and Spain) or endemic goitres [47,48,71,72]. Moreover, the simple fact that someone is under levothyroxine therapy does not indicate if the hypothyroidism is indeed controlled, and the connected cause of hypothyroidism (primary or secondary) and the underlying mechanisms (for example, ATDs, post-thyroidectomy, post-medication, endemic goitres, etc.) are not reflected by the assessment of taking daily thyroxine.

#### 3.1.4. LP/OLP and the Co-Presence of Thyroid Nodules/Malignancy

Thyroid nodules represent the most frequent endocrine condition, affecting 2–6% of the population according to clinical evaluations, with up to one third of adults, based on anterior neck ultrasound assessments, and up to two thirds, as shown by the autopsy investigations in some cohorts [73,74,75]. Whether subjects confirmed with LP/OLP are prone to develop this pathology or whether they represent an incidental finding due to consistent thyroid screening in these patients is yet an open topic [76].

Two already cited case-control studies showed, on the one hand, a higher prevalence of thyroid nodules in 193 persons with OLP versus 163 controls (24.47% versus 20.24%, *p* = 0.004) [58] and, on the other hand, a lower prevalence of this endocrine condition in OLP patients when compared to controls according to another study (10.6% versus 49.6%, *p* < 0.05) [57]. Of note, one single study showed a similar rate of thyroid nodules in OLP and OLL individuals (at 24.47% versus 22.76%, *p* = 0.156) [58]. Moreover, thyroid cancer, estimated as the most frequent endocrine cancer with an increasing incidence amid the modern era [77,78,79], was studied according to one cohort published in 2020, and a similar prevalence in OLP persons versus controls (1.9% versus 1.1%, *p* > 0.05) was identified [57] (Table 1).

Overall, the existing data are not convincing yet to reflect a higher risk of developing thyroid nodules in the LP/OLP population, and future research in terms of thyroid nodule screening (for instance, by using anterior neck ultrasounds), is necessary according to larger, multicentre studies. The massive epidemiologic impact of thyroid nodules in adults might impair an adequate interpretation regarding an increased risk of associating LP/OLP. Also, experimental data are mandatory to highlight the common pathogenic traits (if any) involving the growth of thyroid nodules and the developing/recurrence of LP/OLP.

### 3.2. Subjects with Thyroid Conditions: LP/OLP Analysis

Otherwise, the interplay between the thyroid profile and LP/OLP may be analysed via searching patients with thyroid anomalies and their potential risk of developing LP/OLP. The level of statistical evidence remains rather low [80,81]; two studies pointed to this aspect in patients with ATDs [80] or those who underwent a thyroidectomy; thus, an examination of the oral cavity was performed before surgery [81].

Specifically, Hirota et al. [80] analysed the prevalence of OLP in individuals already diagnosed with ATDs; among the 52 patients (HT/GD ratio of 31/21), 8/52 of them presented oral lesions, but only one GD case had OLP; thus, a prevalence of 1.9% amid ATDs was confirmed [80]. Another cross-sectional study in Italy included 125 patients (74/125 individuals were confirmed with either HT—58/74—or GD—16/74; female/male ratio of 70/4; average age of 47 ± 15.2 years) versus 51 controls (female/male ratio of 42/9; average age of 54.6 ± 11.5 years) with non-autoimmune thyroid conditions such as goitres. All these patients underwent a thyroidectomy (with pre-operatory meticulous investigation of oral mucosa) and a histological exam followed by post-operatory levothyroxine supplementation. A total of 1/58 patients with HT had OLP (with biopsy confirmation); thus, an OLP prevalence of 1.3% amid HT was identified [81]. Yet, no association between HT/ATD and OLP, nor between levothyroxine dosage and OLP, was found [81], contrary to prior data on different populations [48]. Of note, the potential bias of these studies might involve the need of a larger population sample [80], longitudinal data [81], as well as the identification of a specific subgroup of patients with endocrine conditions that might be at a higher risk of LP/OLP.

## 4. Discussion

### 4.1. Study-Based Investigation

Overall, among the data we could find, six studies (five case controls and one cross-sectional) confirmed an association between thyroid dysfunctions (exclusively hypothyroidism) and LP/OLP [43,44,45,46,47,48]. Of note, only one study particularly addressed the cutaneous type of LP, with its main data being focused on OLP [45]. The sample size of the population diagnosed with LP/OLP varied from small sizes of 12–14 persons to large cohorts of more than 1500 individuals. Hypothyroidism prevalence in OLP individuals was identified as follows: OLP patients (N = 76) versus controls (N = 98) with a prevalence of almost 30% [43], while another study enrolled the same type of populations (108 versus 657) but identified higher rates (50% versus 30%) [46]; OLP (N = 14) versus a general population (N = 890) with a similar rate of 30% with regard to reduced blood thyroid hormone concentrations [44]. Moreover, a higher rate of levothyroxine replacement was identified among an OLP group (N = 1611) versus controls (N = 1615) of 10% versus 2.5% [47]. The odds ratio of treated hypothyroidism amid OLP individuals was of 2.99 and 2.19, respectively (95% confidence interval of 2.03–4.44 and of 1.03–4.09, respectively) [47,48]. Interestingly, the presence with hypothyroidism was confirmed to be associated with a milder OLP phenotype in two studies [44,47]. A single cohort revealed a similar prevalence of hypothyroidism in LP versus non-LP subjects [44].

A sneak peek into the non-confirmatory studies (only on OLP, not the cutaneous types that we could identify), as we may call them since a standard correlation between thyroid dysfunction and OLP/LP was not identified, revealed four more studies (one transversal and three case controls) [49,50,51,52] and a new input from a previously mentioned one [47]. These data turned out to show a similar prevalence of hypothyroidism among OLP individuals versus controls, 102 versus 102 subjects [50], 192 versus 163 subjects [51], 523 versus 523 subjects [52], as well as hyperthyroidism among 585 OLP subjects versus 10,441 controls [49] and 192 versus 163 subjects, respectively [51]. Notably, when it comes to statistical significance, a single (relatively large) cohort showed that subjects with OLP actually had a lower prevalence of hypothyroidism versus controls (1% versus 4%), and this seemed to be an isolated result of such a profile [49].

Positive autoimmunity in LP/OLP patients was confirmed in eight studies [43,47,49,51,53,56,59,64] according to our sample-based study, with four of them being the same from the thyroid dysfunction analysis perspective [43,47]; overall, two were retrospective, three were transversal, and three were case-control studies. The sizes of the cohorts enrolling subjects confirmed with the mentioned dermatologic conditions varied, for instance, with 619 persons with LP according to a single study (n = 1/8) [53] and with 76, 92, 105, 108, 192, 247, and 585 patients (a total of 1405) with OLP, respectively [43,49,51,56,64]; notably, the largest control group was of 10,441 individuals [49].

We noticed four clusters of approaches with respect to the autoimmunity in LP/OLP subjects: an analysis of HT/ATD prevalence (regardless of the diagnosis, this was based on antibody concentrations and/or thyroid ultrasound characteristics which are both feasible in daily endocrine practice [82]); considerations over the specific antibody levels; sex-related features since females are more prone to positive autoimmunity; and associations (if any) with the clinical aspects of LP/OLP, including types and severity scores [43,47,51,53,56,64]. HT prevalence in OLP groups versus controls (specifically, 76 versus 98 subjects, 585 versus 10,441 subjects, and 192 versus 163 subjects) was statistically significantly higher, as follows: 19% versus 5% [43]; 12% versus 6% [49]; and 20% versus 9.8% [51]. A single study addressing LP (N = 619) found a 12% rate of ATDs and that females are more affected than males (17% versus 5%) [53]; another study on OLP (N = 247) showed a HT rate of 39%, with women, again, being more frequently involved (46% versus 19%) [56]; the rate of positive serum TPOAb (39% versus 19%) or TgAb levels (25% versus 6%) was confirmed to be higher in women compared to men [56]. While one study did not confirm a correlation between OLP-associated clinical elements and severity and antibody values against the thyroid [56], another showed that a positive TPOAb level is more often assessed in erosive than non-erosive OLP (68% versus 33%) [59]. Just the reverse, one cohort found no correlation between OLP and TPOAb, while OLP subjects had a statistically significantly lower rate of positive TPOAb versus controls (9% versus 15%) [46].

Five case-control studies [47,48,50,71,72] addressed the issue of levothyroxine replacement for a prior endocrine condition in patients that were diagnosed with OLP (no study on LP was identified according to our methods); three of them [47,71,72] confirmed a higher rate of this treatment in OLP (at 8.9%, 9.7%, and 10.6%) versus controls, while the other two did not reach statistical significance [48,50]. When it comes to the thyroid nodule/cancer prevalence in OLP patients, the data are currently limited [57,58], and further larger studies are necessary.

Overall, the strength of the current work is represented by the complexity of these data coming from both dermatologic and endocrine domains; this sample-based analysis is one of the most complex of this type, as far as we are aware. The modern medical era, while being focused on an individual approach, will bring a more specific frame regarding the thyroid profile in LP/OLP or, quite the other way, regarding the LP/OLP subgroup among subjects diagnosed with various thyroid anomalies.

### 4.2. Connecting the Dots between LP/OLP and Thyroid Anomalies

#### 4.2.1. Immune and Autoimmune Elements

Regarding LP/OLP and various thyroid conditions that have been mentioned, a complex puzzle of clinical, biochemical, molecular, pathologic, hormonal, metabolic, genetic, infectious, and other environmental elements is yet to be clarified. In this context, OLP seems to be a more important candidate than cutaneous LP when placed in relationship with thyroid entities, particularly with respect to the immunological processes [83,84,85,86]. A cellular-mediated immunity, triggered by internal or external factors, leads to the generation of TNF-α, IFN-γ, and associations between keratinocytes, T-cells, and antigen-presenting cells [83,87]. The upregulation of T-helper 1-derivate cytokines is crucial at initial OLP occurrence; the genetic variation in cytokines appears to determine whether lesions manifest alone in the oral cavity (linked to IFN-γ) or in both the oral cavity and the skin (linked to TNF-α) [88].

Generally, the term “lichen” refers to a papule that has a flat top, while “lichenoid” describes a morphology that resembles LP. Histologically, the term “lichenoid” refers to a specific pattern of pathological alterations characterized by the presence of a band-like infiltration, mostly composed of mononuclear cells, which obscures the interface between the dermis and epidermis [89]. In the area of dermatology, a multitude of skin disorders are often referred to as “lichen” or “lichenoid”, other than the aforementioned LP; of note, we identified three noted studies addressing the issue of OLL subgroups [43,47,51].

The histological characteristics of OLP and HT suggest that the pathogenic conundrum of both conditions is significantly influenced by the cell-mediated immune response, as they are characterized by the presence of inflammatory infiltration, lymphocyte activation, etc. [90,91]. The pathological report in OLP often includes lymphocyte infiltration, mostly consisting of T lymphocytes, in a band-like pattern underneath the epithelial layer; additionally, basal epithelial cells exhibit liquefaction degeneration and hyper-parakeratosis [92]. HT is histologically distinguished, as well, by the presence of lymphocyte infiltration and fibrosis in the thyroid gland [93].

Moreover, it seems that CD8+ T-cells are implicated in the pathogenesis of both disorders. In OLP, keratinocyte apoptosis is induced by T lymphocyte-mediated cytotoxicity; the majority of T lymphocytes next to the injured basal keratinocytes are CD8+ T-cells which may directly interact with major histocompatibility complex-1 on keratinocytes, leading to chemokine release and other immune cell recruitment, finally causing the formation of OLP lesions [94]. In HT, CD8+ cytotoxic T-cells release perforin and granzyme, thus causing the deterioration of follicular thyroid cells, and, ultimately, it culminates with a cell dysfunction in terms of hypothyroidism [95].

Another common pathogenic factor of both diseases might be the chemokines profile, especially, CXCL10, which is produced and released by several cell types, such as fibroblasts, endothelial cells, keratinocytes, and thyroid follicular cells [96]. By binding to CXCR3, CXCL10 plays a role in many autoimmune illnesses [97]. The epithelial layer of OLP lesions presents a notably elevated expression of CXCL10 and CXCR3 compared to normal tissues [98,99]. The levels of CXCL10 were found to be higher in the blood and tissues of people with ATDs [92,99].

Traditional and modern data connected various endocrine diseases, including those related to the thyroid gland, to skin and mucosal conditions. In ATDs the presence of antibodies against the thyroid might display detrimental cutaneous effects, too. The prevalence of skin anomalies is considerably higher among individuals with thyroid disorders versus general populations [100,101]. Furthermore, the presence of thyroid antibodies in individuals with HT promotes the initiation of autoimmune reactions specific to the oral mucosa, resulting in OLP, while the positive correlation between the severity of OLP lesions and TPOAb levels has been highlighted by some studies, as pointed out above (yet, not all of them) [56,59,64].

Whether high TSH directly acts at the level of LP/OLP lesions via its receptor as a promotor of the lesions in cases with increased TSH (as seen in primary hypothyroidism) is still unclear. For instance, we mention one study from 2020 that analysed the immunohistochemistry expression of TSH and TSH receptors from oral mucosal biopsies. The study included 38 patients out of whom 14 had OLP and HT, 14 had only OLP, and 10 were healthy controls. However, both stains were found negative [102], and a larger exploration of this potential link is mandatory.

#### 4.2.2. Genetic Factors

The polymorphic nature of the HLA system serves as a pivotal genetic determinant in the pathogenesis of numerous diseases, including HT and OLP [103]. An association between HT and HLA-DR3 alleles in Caucasian individuals has been reported [104], while HLA-DR3 has been linked to the erosive variant of OLP [105], respectively. HLA-DPDQDR was connected to palmoplantar LP [106]. Delving into HLA links with both HT and LP might bring to the surface unexpected common loops which are yet to be clarified [107,108,109].

#### 4.2.3. Environmental Elements

Several environmental factors have been implied in the development of OLP and ATDs, such as smoking, vitamin D exposure, infectious diseases, particularly, of viral types, nutrient status, etc. [5,110,111,112,113,114].

##### Smoking

Some data suggest that smoking might be protective against thyroid blocking antibodies (in HT) while being a clear negative contributor to GD onset and progression [115,116]. Genetic predisposition accounts for 80% in GD individuals while environmental factors represent 20%, smoking being a mostly important one, including for thyroid eye disease [117]. On the other hand, smoking also influences the features of dendritic cells which play a crucial role in the activation and development of T cells amid LP [115,116,117].

##### Nutrients

Among the micronutrients that should be taken into consideration, we note that zinc and selenium have been involved the development of ATDs and associated thyroid dysfunctions, as well as OLP/LP, while, according to our methods, no studies were identified that synchronously addressed these aspects. The skin is recognized as the third most zinc-rich organ, and there is a notable association between its deficiency and several dermatologic disorders [118]. In recent times, there has been a notable increase in the number of clinical studies addressing zinc usage as a treatment for a wide range of cutaneous problems, including viral warts, acne, and cutaneous ulcers [119]. This element displays an essential role in facilitating the repair and regeneration of epithelial tissues, also enhancing the local defence mechanisms, hence resulting in a decrease in both inflammation and bacterial proliferation; it also serves as an intermediate in several transcription factors, perhaps emphasizing its regenerative effects [120]. Serum zinc levels have been found to be lower in subjects with oral mucosa lesions or those with erosive OLP [121,122]. As far as the thyroid is concerned, zinc contributes to the pituitary TSH production. It displays a fundamental role in the conversion of thyroxine to triiodothyronine, potentially influencing the levels of circulating thyroid hormones. Zinc supplementation could be beneficial in regulating the serum concentrations of thyroid hormones [123,124], while its levels are associated with thyroid volumes in children and youth across life spans [125,126].

A further exploration of the micronutrient status in relationship with ATD-LP domains is expected, and the theoretic rationale of both conditions also involves a collateral element, namely, selenium [127,128]. Selenium exerts effects by counteracting oxidative stress, therefore decelerating the aging process. Additionally, it hinders viral infections and plays significant roles in chemoprevention by participating in DNA regulation. Furthermore, selenium has metabolic features in the human body, including thyroid hormone metabolism and modulation of the immune system [129]. Since LP represents a chronic inflammatory, immune-mediated disease, the use of selenium seems rational (either with topic applications or systemic administration) [130]. This nutrient might combat both the oxidative stress and the DNA damage that have been proven in LP lesions [131,132]. Selenium might reduce thyroid antibody levels in HT through the upregulation of active T-cells [133]; a low level of selenium might activate Th1/Th2 effectors, hence augmenting immunological responses [134]. While selenium is recommended for mild to moderate thyroid eye disease in GD, some data suggest for it to be used in combination with levothyroxine in HT-associated hypothyroidism [135,136].

##### Vitamin D

Vitamin deficiency represents a major medical and surgical chapter which has not only been connected to the pathogenic traits of LP/OLP but also of ADTs. A total of 80% of the circulating 25-hydroxyvitamin D is derivate from sun exposure via skin synthesis, while the remaining 20% mainly comes from digestive origins. The final step of activation relates to the renal site under the “umbrella” of adequate levels of the parathyroid hormone. The activated vitamin D is displayed all over the human body, and many of the physiological processes at the skin and thyroid glands are connected to its normal levels [137,138,139]. Some authors identified lower levels of 25-hydroxyvitamin D in patients with LP/OLP [140,141]. On the hand, vitamin D replacement in cases with ATDs might improve the outcome, but it does not overcome the standard approach [142,143].

##### Infectious Diseases

Among the potential viral contributors or comorbidities in both ATDs and LP/OLP, hepatitis C has been described with a higher prevalence in certain subgroups. A dual direction with respect to each of these clusters of endocrine and dermatologic conditions should be noted. Among these mechanisms, we mention the fact that viral RNA was found in the oral mucosa and skin. Common autoantibodies have been identified in the serum of LP patients with positive infections. Moreover, the virus exhibits lymphotropism, a phenomenon characterized by its association with certain T-cell responses and with the proliferation of B lymphocytes [144,145,146]. Similarly, autoimmune hypothyroidism was found to be more frequent in patients with hepatitis C than in non-infected individuals; both hepatocytes and lymphocytes demonstrated an elevated production of IFN-γ and IFN-γ inducible chemokines, namely, CXCL10, while ATD patients have elevated concentrations of CXCL10, particularly when accompanied by hypothyroidism [147,148,149].

Recently, the COVID-19 pandemic has placed numerous multidisciplinary conditions in relationship with a new cause, namely, the viral infection itself [150,151]. Some reports showed an exacerbation of LP following coronavirus-associated diseases or even the vaccination against the virus [152,153,154,155], while others highlighted similar data on ATD exacerbation (apart from subacute thyroiditis which is a direct, viral consequence) [156,157,158]. Of note, coronavirus infection might trigger thyroid eye disease in GD or exacerbate TH in previously diagnosed (endocrine) patients, since the virus might interfere with the immune and autoimmune systems as well as the hormonal panel [159,160,161]. This is why we suggest that COVID-19 might represent a potential common trigger for both categories of diseases by sharing signal transduction and pathogenic loops, and this circumstantial evidence (so far) should be explored in the future.

##### 4.2.4. (Non-Thyroid) Hormonal and Metabolic Potential Contributors to LP/OLP and Thyroid Conditions

Prior research has shown that vitamin D might interfere with the synthesis of IFN-γ and IL-1β within the epithelial cells. Furthermore, vitamin D deficiency may potentially contribute to the development of OLP progression, as it has a pivotal function in the regulation of inflammation and cytokine synthesis. Additionally, it exhibits antibacterial and immunomodulatory properties, while demonstrating a suppressive impact on T-helper 1 immune response. Potential positive effects of its supplementation in OLP patients have been found [162,163,164].

Additionally, a vitamin D-associated immunosuppressive role has been acknowledged in HT, decreasing pro-inflammatory cytokine production, therefore inhibiting the immunological response mediated by cytokines and suppressing the expression of HLA class II genes in the thyroid, leading to the inhibition of lymphocyte proliferation and inflammatory cytokines release [165,166,167,168]. Hipovitaminosis D leads to the proliferation and differentiation of B cells into plasma cells, ultimately resulting in increased levels of immunoglobulin G and E secretion, thyroid cell damage, and HT triggering; its replacement in “D”-efficient patients might help the outcome of HT and BD [169,170,171].

Menopause-related oestrogen deficiency is described as a contributor to atrophic modifications in the oral mucosa which has oestrogen receptors and a potential exacerbation of autoimmune manifestations, including pemphigus vulgaris, and OLP has been revealed [172,173]. On the other hand, the post-menopausal status might add to the aging effect both causing a thyroid function decline [174]. Of note, the studies from Table 1 specifically followed adult populations. While a sex-related distribution of both the conditions has been mentioned, the data regarding the impact of menopausal status has not been identified in any of them.

The release of IL-2, IL-4, IL-6, IL-10, and TNF-α cytokines by the Th1 cells involved in the pathogenesis of LP has been implicated in the development of metabolic syndrome too; a positive correlation between this condition and LP highlights the issue of potential metabolic connections in LP/OLP patients [175]. Notably, adults with hypothyroidism might be affected by a higher number of metabolic complications as well [176,177]. As mentioned earlier, the IL-derivate panel is connected to LP/OLP and ATDs, for example, IL-8 in erosive OLP [59,60].

### 4.3. From Key Findings to Current Limits of This Topic and Future Research

Potential clinical implications of the current data with respect to multiple connections between thyroid conditions of various types and the presence of LP and/or OLP involve, firstly, awareness that the same population group might be affected by the dermatologic and endocrine diseases (simultaneously or not, across life spans); secondarily, some patients with LP/OLP might require a detailed hormonal evaluation and consecutive follow-up if thyroid anomalies are identified as, for example, the need of lifelong levothyroxine requirement or the increased risk of developing hypothyroidism in individuals with positive blocking antibodies; and, finally, we should consider that autoimmune thyroiditis and LP/OLP might belong to the same cluster of general autoimmune diseases, and other organs might be targeted, as well, under these circumstances (Figure 1).

As limitations of the current work, we mention the design in terms of a narrative review (without a specific evaluation of the quality of each included study) according to a single database research which, however, afforded us a flexible approach of various sub-chapters underlying different levels of statistical evidence amidst analysing the LP/OLP and thyroid connections. As mentioned, some important topics such as thyroid dysfunction and positive autoimmunity in patients diagnosed with LP/OLP are linked to both confirmatory and non-confirmatory studies. This is why a clear conclusion still represents an open issue at this point. Moreover, a lot of the data actually involved individuals with OLP, rather than LP, thus pinpointing to the fact that two distinct subgroups of individuals should be taken into consideration when analysing the thyroid profile, a specific issue which is yet to be determined.

Multiple clinical data are not convincing yet, as seen with regard to the thyroid nodules and malignancies in OLP/LP. Further pathogenic studies will bring an expansion to the current knowledge and understanding of common traits as discussed above. Awareness of potential thyroid damage is necessary in LP/OLP and interventional (longitudinal) studies, and it is mandatory to display the long-term outcome in patients diagnosed with both conditions and potential interferences of specific therapies. Moreover, inter-disciplinary algorithms involving the panel of baseline assessments and long-term follow-ups of subjects suffering from LP/OLP and thyroid anomalies should help clinicians with different backgrounds.

## 5. Conclusions

With regard to LP/OLP and thyroid anomalies via the clinical studies that were found according to our methods, we identified six confirmatory and five non-confirmatory studies regarding the correlation between thyroid dysfunction and LP/OLP. Eight studies addressed the issue of positive thyroid autoimmunity amid LP/OLP and five introduced the parameters of LP/OLP patients under prior levothyroxine replacement, while two cohorts displayed data with respect to thyroid nodules/cancer. Overall, we mention several main aspects as practical points for multidisciplinary practitioners: OLP rather than LP requires thyroid awareness; when it comes to the type of thyroid dysfunction, mostly, hypothyroidism should be taken into consideration; female patients are more prone to be affected with respect to the ATD association; a potential higher ratio of OLP subjects taking levothyroxine was found; thus, a good collaboration with an endocrinology team is mandatory; so far, OLP individuals do not confirm to associate with a higher risk of thyroid nodules or thyroid cancer. Future research is mandatory to identify the underlying common pathogenic pathways and to highlight the connections between the thyroid anomalies and LP/OLP, the real epidemiologic impact of these connections, as well as the practical recommendations to be addressed amid daily practices. Whether a subgroup of individuals is specifically affected by these dermatologic and endocrine diseases is yet to be detected.

## Figures and Tables

**Figure 1 biomedicines-12-00077-f001:**
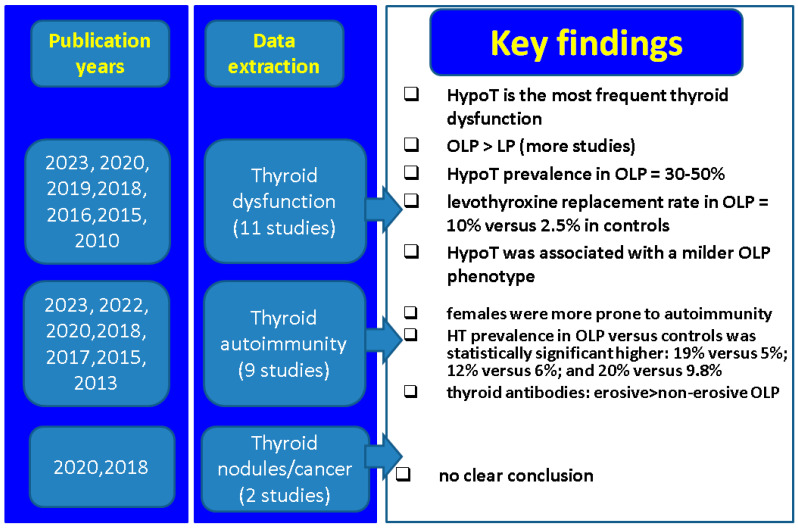
Key findings across mentioned studies with respect to LP/OLP and thyroid connections [43,44,45,46,47,48,49,50,51,52,53,56,64,71,72]. Abbreviations: LP = lichen planus; OLP = oral LP, HypoT = hypothyroidism; HT = Hashimoto’s thyroiditis.

**Table 1 biomedicines-12-00077-t001:** The sample-based analysis of previously published studies in the field of patients diagnosed with LP/OLP that were checked for different forms of thyroid conditions; the studies from each subsection are displayed from the most recent to the oldest publication date [43,44,45,46,47,48,49,50,51,52,53,56,64,71,72].

Authors Year of Publication Reference Number	Type of Study Country	Studied Subgroups	Outcomes
LP/OLP and thyroid dysfunction: confirmatory studies			
Piloni 2023 [43]	Retrospective, case-control study (#) Italy	N = 307 patients 158 (51.47%) F + 149 (48.53% M) (age > 18 y; range: 49–75 y, mean age of 63 y) N1 = 76 OLP patients N2 = 133 OLL patients N3 = 98 controls	Patients with hypothyroidism: N1′ = 17/76 patients (22.37%) N2′ = 19/133 patients (14.29%) N3′ = 7/98 patients (7.14%) N1′ vs. N3′: *p* = 0.0059 N2′ vs. N3′: *p* = 0.09
Amato-Cuartas 2019 [44]	Cross-sectional study Colombia	N = 860 patients N1 = 14 OLP patients (representing 1.6% of N, mean age of 59.6 y, F = 11/16)	Patients with hypothyroidism: N′ = 34/860 patients (3.9%) N1′ = 5/14 patients (35.7%) N′ vs. N1′: *p* = 0.0001
Kumar 2019 [45]	Case-control study India	N = 75 LP patients (adults with at least 3-month disease duration) 43 (57.33%) F and 32 (42.66%) M age: 16–45 y (mean age of 35.2 y) 17.3% with metabolic syndrome N1 = 12 OLP patients N2 = 75 controls	Patients with hypothyroidism: N′ = 8/75 patients (10.76%) N1′ = 4/12 patients (33%) N2′ = 3/75 patients (4%) N′ vs. N1′: *p* = 0.005 N′ vs. N2′: *p* = 0.117
Robledo-Sierra 2018 [46]	Case-control study Sweden (*)	N = 110 OLP+/LT_4_− 99 (90%) F (mean age of 65.6 y) N1 = 657 healthy controls	Patients with hypothyroidism: N′ = 62/108 patients (58.1%) N1′ = 21/58 patients (37.3%) N′ vs. N1′: *p* = 0.0179
Robledo-Sierra 2015 [47]	Case-control study Sweden (#,#)	N = 1611 OLP patients (data collection from general population: standardized registration method) N1 = 1615 controls	Patients with hypothyroidism: N′ = 180/1611 patients (11%) N1′ = 40/1615 patients (2.5%) multivariate OR = 2.99 (95% CI: 2.03–4.44; *p* < 0.0001)
Siponen 2010 [48]	Retrospective case-control study Finland (#,#,#)	N = 222 OLP/OLL patients N1 = 222 controls	Patients with hypothyroidism: N′ = 21/222 patients (10%) N1′ = 10 /222 patients (5%) OR = 2.19 (95% CI: 1.03–4.90)
LP/OLP and thyroid dysfunction: non-confirmatory studies			
Tang 2020 [49]	Cross-sectional study China (#,#,#,#)	N = 585 OLP patients 167 (87.97%) F + 23 (12.1% M) (age > 18 y; mean age of 52.85 ± 13.15 y) N1 = 10.441 controls	Patients with hypothyroidism: N′ = 7/585 patients (1.2%) N1′ = 430/10.441 patients (4.1%) N′ vs. N1′: *p* < 0.05 Patients with hyperthyroidism: N″ = 6/585 patients (1%) N1″ = 48/10.441 patients (0.5%) N” vs. N1”: *p* > 0.05
Kats 2019 [50]	Retrospective, case-control study Israel (#,#,#,#,#)	N = 102 OLP patients 70.6% F (mean age of 55.7 ± 13.2 y) N1 = 102 controls	Patients with hypothyroidism: N′ = 13/102 patients (12.7%) N1′ = 10/102 patients (9.8%) N′ vs. N1′: *p* = 0.65
Zhou 2018 [51]	Case-control study China (#,#,#,#,#,#)	N = 192 OLP patients 75.5% F + 24.45% M (mean age of 49.53 ± 9.93 y) N1 = 123 OLL patients N2 = 163 controls	Patients with hypothyroidism: N′ = 10/192 patients (5.2%) N1′ = 6/123 patients (4.87%) N′ vs. N1′: *p* = 0.565 N2′ = 8/163 patients (4.9%) N′ vs. N2′: *p* = 0.225 Patients with hyperthyroidism: N″ = 3/192 patients (1.56%) N1″ = 1/123 patients (0.81%) N″ vs. N1″: *p* = 0.845 N2″ = 1/163 patients (0.61%) N″ vs. N2″: *p* = 0.402
Lavaee 2016 [52]	Retrospective, comparative study Iran	N = 523 OLP patients 387(74%) F + 136 (26.2%) M (age > 18 y; mean age of 48.43 ± 13.82 y) N1 = 523 controls	Patients with hypothyroidism: N′ = 35/523 patients (6.7%) N1′ = 21/523 patients (4%) OR = 1.714 (95% CI: 0.984–2.987) Chi-square test showed no significant association between hypothyroidism and OLP
LP/OLP and positive thyroid autoimmunity: confirmatory studies			
Piloni 2023 [43]	Retrospective, case-control study (#)	N = 307 patients N1 = 76 OLP patients N2 = 133 OLL patients N3 = 98 controls	Patients with HT: N1′ = 15/76 patients (19.74%) N2′ = 15/133 patients (11.28%) N3′ = 5/98 patients (5.1%) N1′ vs. N3′: *p* = 0.005 N2′ vs. N3′: *p* = 0.1075
Anttonen 2023 [53]	Retrospective study Northern Finland	N = 619 LP patients 58.3% F (mean age of 54.2 y) N1 = 258 LP males (41.7%) N2 = 361 LP females (58.3%)	Patients with ATD: N′ = 12.44% patients N1′ = 5% patients N2′ = 17.7% patients N1′ vs. N2′: *p* < 0.001
Zhang 2022 [56]	Cross-sectional study China	N = 247 OLP patients 186 (75.3%) F + 61 (24.7%) M (mean age of 45.21 ± 12.72 y) N1 = 61 OLP males (24.70%) N2 = 186 OLP females (75.30%)	Patients with HT: N′ = 98/247 patients (39.67%) N1′ = 12/61 patients (19.67%) N2′ = 86/186 patients (46.24%) N1′ vs. N2′: *p* = 0.000
Tang 2020 [49]	Cross-sectional study (#,#,#,#)	N = 585 OLP patients N1 = 10,441 controls	Patients with HT: N′ = 71/585 patients (12.1%) N1′ = 638/10.441 patients (6.1%) N′ vs. N1′: *p* < 0.05
Zhou 2018 [51]	Case-control study (#,#,#,#,#,#)	N = 192 OLP patients N1 = 123 OLL patients N2 = 163 controls	Patients with HT: N′ = 40/192 patients (20.83%) N1′ = 23/123 patients (18.69%) N′ vs. N1′: *p* = 0.011 N2′ = 16/163 patients (9.81%) N′ vs. N2′: *p* = 0.000
Alikhani 2017 [59]	Case-control study Iran	N = 92 OLP patients 34 (74%) F (mean age of 45 y, ranges: 17–68) N1 = 44 EOLP patients N2 = 48 non-EOLP patients	Patients with TPOAb + ve status: N1′ = 30/44 patients (68%) N2′ = 33/48 patients (33%) N1′ vs. N2′: *p* = 0.023
Robledo-Sierra 2015 [47]	Case-control study (#,#)	N = 108 OLP patients N1 = 40 controls	Patients with HT: N′ = 19/108 patients (22.1%) N1′ = 1/40 patients (2.5%) N′ vs. N1′: *p* = 0.0037
Lo Muzio 2013 [64]	Cross-sectional study Italy	N = 105 OLP patients N′ = 15/105 patients with OLP + HT (14/15 F + 1/15 M; age ranges: 33–63 y)	Patients with HT: N′ = 15/105 patients (14.3%) N′ vs. general population: *p* < 0.0003 (general population with a HT-related hypothyroidism prevalence of 1%)
LP/OLP and positive thyroid autoimmunity: non-confirmatory studies			
Robledo-Sierra 2018 [46]	Case-control study (*)	N = 110 OLP+/LT_4_− N1 = 657 controls N2 = 108 OLP+/LT_4_+ N3 = 58 OLP−/LT_4_+	Increased levels of TPOAb: N′ = 9/110 patients (9.4%) N1′ = 99/657 patients (15%) N′ vs. N1′: *p* = 0.024
LP/OLP and levothyroxine replacement for an endocrine condition: confirmatory studies			
Garcia-Pola 2016 [71]	Case-control study Spain	N = 215 OLP patients N1 = 215 controls	Patients with levothyroxine usage: N′ = 21/215 patients (9.7%) N1′ = 7/215 patients (3.3%) N′ vs. N1′: *p* = 0.006
Robledo-Sierra 2015 [47]	Case-control study (#,#)	N = 1611 OLP patients N1 = 1615 controls	N′ = 170/1611 patients (10.6%) N1′ = 40/1615 patients (2.5%) N′ vs. N1′: *p* < 0.0001
Robledo-Sierra 2013 [72]	Case-control study Sweden	N = 956 OLP patients N1 = 1029 controls	Patients with levothyroxine usage: N′ = 85/956 patients (8.9%) N1′ = 26/1029 patients (2.5%) N′ vs. N1′: *p* < 0.001
LP/OLP and levothyroxine replacement for an endocrine condition: non-confirmatory studies			
Kats 2019 [50]	Retrospective, case-control study (#,#,#,#,#)	N = 102 OLP patients N1 = 102 controls	Patients with levothyroxine usage: N′ = 13/102 patients (12.7%) N1′ = 10/102 patients (9.8%) N′ vs. N1′: *p* = 0.659
Siponen 2010 [48]	Retrospective case-control study (#,#,#)	N1 = 222 OLP/OLL patients N = 222 controls	Patients with levothyroxine usage: N′ = 21/222 patients (10%) N1′ = 11/222 patients (5%) OR = 0.94 (95% CI: 0.37–5.2)
LP/OLP and thyroid nodules/cancer			
Tang 2020 [49]	Cross-sectional study (#,#,#,#)	N = 585 OLP patients N1 = 10,441 controls	Patients with thyroid nodules: N′ = 62/585 patients (10.6%) N1′ = 5.183 /10,441 patients (49.6%) N′ vs. N1′: *p* < 0.05 Patients with thyroid cancer: N″ = 11/585 patients (1.9%) N1″ = 117/10.441 patients (1.1%) N” vs. N1”: *p* > 0.05
Zhou 2018 [51]	Case-control study (#,#,#,#,#,#)	N = 192 OLP patients N1 = 123 OLL patients N2 = 163 controls	Patients with thyroid nodules: N′ = 47/192 patients (24.47%) N1′ = 28/123 patients (22.76%) N′ vs. N1′: *p* = 0.156 N2′ = 33/163 patients (20.24%) N′ vs. N2′: *p* = 0.004

Abbreviations: ATD = autoimmune thyroid disease; CI = confidence interval; EOLP = erosive oral lichen planus; F = female; HT = Hashimoto’s thyroiditis; LP = lichen planus; M = male; N = number of patients; OLP = oral lichen planus; OR = odds ratio; OLP+/LT_4_− = patients with OLP and no history of thyroid disease and/or levothyroxine therapy; OLP+/LT_4_+ = patients with OLP and levothyroxine therapy; OLP−/LT_4_+ = patients under levothyroxine replacement without the diagnosis of OLP; OLL = oral lichenoid lesion; *p* = *p*-value; vs. = versus; TPOAb = anti-thyroperoxidase antibodies; y = years; (#); (#,#); (#,#,#); (#,#,#); (#,#,#,#,#); (#,#,#,#,#,#); (*)—the same study is introduced at different sections based on the original outcomes.

## Data Availability

Not applicable.

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
