# Peer review of "Crossroads between Skin and Endocrine Glands: The Interplay of Lichen Planus with Thyroid Anomalies"

_biomedicines, 2023, doi:10.3390/biomedicines12010077_

Round 1

Reviewer 1 Report

Comments and Suggestions for Authors

Overall, the manuscript provides a comprehensive overview of the interplay between lichen planus (LP) and thyroid conditions (TC) from both dermatologic and endocrine perspectives. The literature review is extensive, and the methodology appears sound. However, some areas need attention for clarification and improvement.

1. The title is clear, but consider adding a subtitle to give a more specific indication of the manuscript's focus.

2. The abstract is detailed but could be condensed for better readability. Important information, such as the key findings and their implications, should be highlighted more explicitly.

3. The manuscript contains a wealth of data, and it might be beneficial to present a summary of the main findings in each section before delving into the details.

4. The discussion is insightful, but it would be helpful to explicitly connect the findings to the broader literature and discuss the potential clinical implications of the observed associations.

5. Address any limitations of the included studies, and discuss how these limitations might impact the overall conclusions drawn from the narrative review. When discussing non-confirmatory studies, it's essential to highlight potential reasons for the conflicting results and encourage further research to reconcile disparities.

6. The conclusion is well-stated but could be more concise. Summarize the key takeaways and their relevance in a few sentences.

7. Ensure that the language is clear and concise, making it accessible to a broad audience. Ensure consistency in terminology throughout the manuscript. Check for typographical errors and formatting inconsistencies.

Specific Comments on Sections:

2.1.4. LP/OLP and the Co-Presence of Thyroid Nodules/Malignancy (Lines 313-328):The paragraph discusses conflicting findings regarding the association between LP/OLP and thyroid nodules. It's essential to emphasize the need for further research and explore potential reasons for the contradictory results.

2.2. Subjects with Thyroid Conditions: LP/OLP Analysis (Lines 340-357): The section describes studies analyzing the interplay between thyroid conditions and LP/OLP. It is well-structured and provides a good summary of the studies. However, it would be beneficial to discuss the limitations of the studies and the need for more robust evidence.

4.2. Connecting the Dots between LP/OLP and Thyroid Anomalies: The section provides an in-depth analysis of immune elements, genetic factors, environmental elements, and hormonal/metabolic contributors. The content is comprehensive but could be condensed for clarity. The discussion on genetic factors, environmental elements, and hormonal/metabolic contributors is informative. However, presenting this information in a more structured manner might enhance readability.

4.2.3. Environmental Elements (Lines 483-489): The section provides a comprehensive overview of environmental factors. However, separating information on smoking, vitamin D, infectious diseases, and nutrients into distinct subsections might enhance clarity.

4.2.4. (Non-Thyroid) Hormonal and Metabolic Potential Contributors (Lines 542-570): The discussion on vitamin D, IL-2, IL-4, IL-6, IL-10, and TNF-α cytokines is insightful. However, connecting these factors explicitly to both LP/OLP and thyroid conditions would strengthen the argument.

**5. Conclusions:**

- The conclusion summarizes the key findings. However, it would be beneficial to restate the main implications for clinical practice and potential avenues for future research.

Overall, the manuscript presents a thorough review of the association between LP/OLP and thyroid anomalies. Addressing the specific comments and recommendations for improvement should enhance the clarity and impact of the manuscript.

Comments on the Quality of English Language

Minor refinements required. Please adhere to clarity of terms and ensure consistency in terminology throughout the manuscript. Check for typographical errors and formatting inconsistencies.

Author Response

Response to Reviewer’ Comments – Reviewer number 1

Dear Reviewer,

Thank you very much for your time and your effort to review our manuscript.

We are very grateful for providing your valuable feedback on the article.

Here is our response and related amendment that has been made in the manuscript according to your review (marked in yellow colour).

…….

Overall, the manuscript provides a comprehensive overview of the interplay between lichen planus (LP) and thyroid conditions (TC) from both dermatologic and endocrine perspectives. The literature review is extensive, and the methodology appears sound.

Thank you very much. We really appreciate it. We addressed the issue of methodology as mentioned below. Thank you.

…….

However, some areas need attention for clarification and improvement.

Thank you very much. We will pinpoint the answer to each of them as follows.

…….

The title is clear, but consider adding a subtitle to give a more specific indication of the manuscript's focus.

Thank you very much. The title has been modified according to the recommendation of the reviewer number 2, as well. We respectfully consider this format to be enough in order to suggest the content of the paper since the data that has been analysed are very different and come from a multidisciplinary background. Thank you. 

…….

The abstract is detailed but could be condensed for better readability. Important information, such as the key findings and their implications, should be highlighted more explicitly.

Thank you very much. We corrected the abstract. Thank you

…….

The manuscript contains a wealth of data, and it might be beneficial to present a summary of the main findings in each section before delving into the details.

Thank you very much. We introduced the main findings in each section as, for example, the followings:

  1. Section 3.1.1. LP/OLP and Thyroid Dysfunction

“Regarding LP/OLP most studies addressed the issue of hypothyroidism, rather than hyperthyroidism, while the dermatologic approach is mainly focused on OLP, not LP in order to address the thyroid function. Overall, ten studies addressed the issue of identifying any type of abnormal thyroid hormones levels in LP/OLP.”

  1. Section 3.1.2. LP/OLP and Positive Thyroid Autoimmunity

“When facing the thyroid autoimmune background in LP/OLP, either in terms of HT or GD, the sample-based analysis cvasi-exclusively pointed HT rather than GD; ATDs were confirmed by the assessment of serum antibodies and sometimes a secondary criterion of thyroid ultrasound features was added, despite not being mandatory in daily practice to sustain the diagnosis itself [42]. Among the eight confirmatory studies we identified, two of them were mentioned above since the thyroid function was analysed, as well [43,47]. An additional study we appreciated as being non-confirmatory in this particular matter. The key finding remains a higher rate of positive thyroid antibodies in terms of HT diagnosis among LP/OLP which sustains a certain level of awareness across dermatological practice. Secondarily, women seem more susceptible to associate positive thyroid antibodies.”

  1. Section 3.1.3. LP/OLP Patients under Levothyroxine Replacement

Under these specifications, we mention that some studies distinctly analysed the status of levothyroxine therapy (as part of the endocrine management, not of the specific approach in LP/OLP) and found discordant results; yet, the pitfall of addressing this insight is represented by the non-homogenous background of underlying thyroid anomalies, and by different levels of controlled/uncontrolled hypothyroidism, as well as, target TSH in accordance to various guidelines depending on the primary endocrine condition [68-70]. Thus, two additional case-control studies [71,72] and a prior cited one…”

“Nevertheless, two other mentioned case-control studies did not confirm these results (also, exclusively in OLP, not in LP)”…

  1. Section 1.4. LP/OLP and the Co-Presence of Thyroid Nodules/Malignancy

“Whether subjects confirmed wit LP/OLP is prone to develop this pathology or they represent an incidental finding due to consistent thyroid screening in these patients is yet an open topic [76]. Two already cited case-control studies showed, on one hand, a higher prevalence of thyroid nodules in 193 persons with OLP persons versus 163 controls (24.47% versus 20.24%, p = 0.004) [58], and, on the other hand, a lower prevalence of this endocrine condition in OLP when compare to controls according to another study (10.6% versus 49.6%, p < 0.05) [57]. Of note, one single study showed a similar rate of thyroid nodules in OLP and OLL (of 24.47% versus 22.76%, p = 0.156) [58].”

Thank you

…….

The discussion is insightful, but it would be helpful to explicitly connect the findings to the broader literature and discuss the potential clinical implications of the observed associations.

Thank you very much. This section has been expanded. Also, we introduced potential clinical implications, a larger panel of limitations, and future development of this current topic.

 “Potential clinical implications of the current data with respect to multiple connections between thyroid conditions of various types and the presence of LP and/or OLP involve firstly awareness that the same population group might be affected by the dermatologic and endocrine diseases (simultaneously or not, across life span); secondarily, some patients with LP/OLP might require a detailed hormonal evaluation and consecutive follow-up if thyroid anomalies are identified as, for example, the need of lifelong levothyroxine requirement or the increased risk of developing hypothyroidism in individuals with positive blocking antibodies; and, finally, we should consider that autoimmune thyroiditis and LP/OLP might belong to the same cluster of general autoimmune diseases and other organs might be targeted, as well, under these circumstances. 

As limitations of the current work we mention the design in terms of narrative review (without a specific evaluation of the quality of each included study) according to a single database research which, however, allowed us a flexible approach of various sub-chapters underlying different levels of statistical evidence amidst analysing the LP/OLP and thyroid connections. As mentioned, some important topics such as thyroid dysfunction and positive autoimmunity in patients diagnosed with LP/OLP are linked to both confirmatory and non-confirmatory studies; that is why a clear conclusion still represents an open issue at this point. Moreover, many data actually involved the individuals with OLP, rather than LP, thus pinpointing the fact that two distinct subgroups of individuals should be taken into consideration when analysing the thyroid profile, a specific issue which is yet to be determined.

Multiple clinical data are not convincing yet, as seen with regard to the thyroid nodules and malignancies in OLP/LP. Further pathogenic studies will bring an expansion to the current knowledge and understanding of common traits as discussed above. Awareness of potential thyroid damage is necessary in LP/OLP and interventional (longitudinal) studies are mandatory to display the long term outcome in patients diagnosed with both conditions and potential interferences of specific therapies. Moreover, inter-disciplinary algorithms involving the panel of baseline assessments and long term follow-up of subjects suffering from LP/OLP and thyroid anomalies should help the clinicians with different backgrounds”.

Thank you

…….

Address any limitations of the included studies, and discuss how these limitations might impact the overall conclusions drawn from the narrative review. When discussing non-confirmatory studies, it's essential to highlight potential reasons for the conflicting results and encourage further research to reconcile disparities.

Thank you very much. We introduced the studies – related limitations as, for instance, the followings:

  1. Section 3.1.1. LP/OLP and Thyroid Dysfunction

“As limitations of the cited studies that might bring additional biases in interpretation  of the association between thyroid dysfunction and LP/OLP, we mention: retrospective type of study with a prior collection of the variables [43,46,48,52]; the parameters were based on a population belonging to a specific geographic area that does not allow a generalization [43,44]; small sample size [44]; interferences of the metabolic issues such as dyslipidaemia and metabolic syndrome in addressing the rate of hypothyroidism and its consequences [45]; the comparison with control group was done according to historical (not real-time) data [49]; the analysis of the etiological types of hypothyroidism was not performed, thus pointing a potential pathogenic clue is not applicable [52]. Moreover, concomitant drugs might falsely supress serum TSH levels such as glucocorticoids administration, and somatostatin analogues like octreotide thus the thyroid panel might not be accurate [50].”

  1. Section 3.1.2. LP/OLP and Positive Thyroid Autoimmunity

 “As limits of these studies, we specify: registry-based collection of parameters [53]; different geographic areas such as Italy [43], Sweden [47], Finland [53], China [49,51,56], and Iran [59]; the evaluation was done at hospital that is why less severe patients of LP who were followed at primary care centres were not included [53]; the inter- and intra-individual fluctuations of the serum thyroid antibodies (which usually do not serve as prognostic markers in daily endocrine practice) [49,51,57,58]. Notably, one correlation does not mean causality, while the association between LP/OLP and ATDs might be regarded amid the general panel of increased autoimmunity risk of these patients and not a specific (tide) connection between LP/OLP and ATDs [43,46,47,49,51,53,56,59,64]. Additionally, the timing of identifying positive thyroid antibodies in relationship with the confirmation of OLP should be studied since these antibodies might activate OLP and then decreased at serum level, thus a cross-sectional analysis might not be conclusive [59]. Also, histological confirmation of HT (which is not mandatory in daily practice, neither it is commonly used) might limit the enrolment of the individuals with HT diagnosis among those already confirmed with LP/OLP [64].”

  1. Section 3.1.3. LP/OLP Patients under Levothyroxine Replacement

“Among the potential limitations of the cited studies, we mention the fact that survey-based data collection with respect to the levothyroxine administration might bring an additional bias [47]. Also, different geographic areas associate distinct rates of thyroxine - requiring hypothyroidism (Sweden, Finland, Spain)or endemic goitre [47,48,71,72]. Moreover, the simple fact that someone is under levothyroxine therapy does not indicate if the hypothyroidism is indeed controlled, neither to say that the connected cause of hypothyroidism (primary or secondary) and the underlying mechanisms (for example, ATDs, post-thyroidectomy, post-medication, endemic goitre, etc.) are not reflected by the assessment of taking daily thyroxine.”

Thank you

…….

The conclusion is well-stated but could be more concise. Summarize the key takeaways and their relevance in a few sentences.

Thank you very much. We adjusted the conclusion. However, we respectfully have to take into consideration the recommendation of the reviewer number 2 in order to add some additional findings. Thank you

…….

Ensure that the language is clear and concise, making it accessible to a broad audience. Ensure consistency in terminology throughout the manuscript. Check for typographical errors and formatting inconsistencies.

Thank you very much. We checked again the text. Thank you

…….

Specific Comments on Sections:

2.1.4. LP/OLP and the Co-Presence of Thyroid Nodules/Malignancy (Lines 313-328): The paragraph discusses conflicting findings regarding the association between LP/OLP and thyroid nodules. It's essential to emphasize the need for further research and explore potential reasons for the contradictory results.

Thank you very much. We expanded the data according to your recommendation, as following:

“Overall, the existing data are not convincing yet to reflect a higher risk of developing thyroid nodules in LP/OLP population and future research in terms of thyroid nodule screening (for instance, by using anterior neck ultrasound) is necessary according to larger, multicentre studies. The massive epidemiologic impact of thyroid nodules in adults might impair an adequate interpretation regarding an increased risk of associating LP/OLP. Also, experimental data are mandatory to highlight the common pathogenic traits (if any) involving the growth of thyroid nodules and the developing/recurrence of LP/OLP.”

Thank you

…….

2.2. Subjects with Thyroid Conditions: LP/OLP Analysis (Lines 340-357): The section describes studies analyzing the interplay between thyroid conditions and LP/OLP. It is well-structured and provides a good summary of the studies. However, it would be beneficial to discuss the limitations of the studies and the need for more robust evidence.

Thank you very much. We introduced the limitations according to your recommendation as follows:

“Of note, the potential bias of these studies might involve the need of a larger population sample [80], longitudinal data [81], as well as the identification of a specific subgroup of patients with endocrine conditions that might be at higher risk of LP/OLP. “

Thank you

…….

4.2. Connecting the Dots between LP/OLP and Thyroid Anomalies: The section provides an in-depth analysis of immune elements, genetic factors, environmental elements, and hormonal/metabolic contributors. The content is comprehensive but could be condensed for clarity. The discussion on genetic factors, environmental elements, and hormonal/metabolic contributors is informative. However, presenting this information in a more structured manner might enhance readability.

Thank you very much. We respectfully mention that the section number 4.2. was introduced to highlight the complexity of the thyroid connections with respect to the field of LP/OLP, while it does not represent the specific topic of the current work. We adjusted it according to your recommendations. Thank you

For instance, we highlight a few adjustments and add, as followings:

Section 4.2.2.

“…respectively; HLA-DPDQDR was connected to palmoplantar LP [106]. Delving into HLA links with both HT and LP might bring to the surface unexpected common loops which are yet to be clarified.“

Added references

  1. Abreu Velez AM, Howard MS, Pereyo N. Palmar and plantar lichen planus: a case report and review of the literature.An Bras Dermatol. 2015;90(3 Suppl 1):175-7. doi:10.1590/abd1806-4841.20153034.
  2. Sernicola A, Mazzetto R, Tartaglia J, Ciolfi C, Miceli P, Alaibac M. Role of Human Leukocyte Antigen Class II in Antibody-Mediated Skin Disorders.Medicina (Kaunas). 2023;59(11):1950. doi: 10.3390/medicina59111950.
  3. Stasiak M, Stasiak B, Zawadzka-Starczewska K, Lewiński A. Significance of HLA in Graves' disease and Graves' orbitopathy in Asian and Caucasian populations-a systematic review. Front Immunol. 2023;14:1256922. doi:10.3389/fimmu.2023.1256922.
  4. Berryman MA, Ilonen J, Triplett EW, Ludvigsson J. Important denominator between autoimmune comorbidities: a review of class II HLA, autoimmune disease, and the gut. Front Immunol. 2023;14:1270488. doi:10.3389/fimmu.2023.1270488.

Section 4.2.3.A. Smoking

“Genetic predisposition accounts for 80% in GD while environmental factors represent 20%, smoking being a mostly important one, including for thyroid eye disease.”

Added reference:

Antonelli A, Ferrari SM, Ragusa F, Elia G, Paparo SR, Ruffilli I, Patrizio A, Giusti C, Gonnella D, Cristaudo A, Foddis R, Shoenfeld Y, Fallahi P. Graves' disease: Epidemiology, genetic and environmental risk factors and viruses. Best Pract Res Clin Endocrinol Metab. 2020;34(1):101387. doi:10.1016/j.beem.2020.101387.

Section 4.2.3.B. Nutrients

Since LP represents a chronic inflammatory, immune-mediated disease, the use of selenium seems rationale (either with topic applications or systemic administration) [130]. The nutrient might combat both the oxidative stress and the DNA damage that have been proven in LP lesions [131,132].

Added references:

Darczuk D, Krzysciak W, Vyhouskaya P, Kesek B, Galecka-Wanatowicz D, Lipska W, Kaczmarzyk T, Gluch-Lutwin M, Mordyl B, Chomyszyn-Gajewska M. Salivary oxidative status in patients with oral lichen planus. J Physiol Pharmacol. 2016 ;67(6):885-894.

Agha-Hosseini F, Mirzaii-Dizgah I, Farmanbar N, Abdollahi M. Oxidative stress status and DNA damage in saliva of human subjects with oral lichen planus and oral squamous cell carcinoma. J Oral Pathol Med. 2012;41(10):736-40. doi: 10.1111/j.1600-0714.2012.01172.x. 

Barikbin B, Yousefi M, Rahimi H, Hedayati M, Razavi SM, Lotfi S. Antioxidant status in patients with lichen planus. Clin Exp Dermatol. 2011;36(8):851-4. doi:10.1111/j.1365-2230.2011.04152.x. 

Section. 4.2.3.C. Vitamin D

Vitamin deficiency represents a major medical and surgical chapter which has been connected to the pathogenic traits of LP/OLP, but, also, of ADTs. 80% of the circulating 25-hydroxyvitamin D is derivate from the sun exposure via skin synthesis while the remaining 20% mainly comes from the digestive origin. The final step of activation relates to the renal site under the “umbrella” of adequate levels of parathyroid hormone. The activated vitamin D is displayed all over the human body and many of the physiological processes at skin and thyroid gland are connected to its normal levels [137-139]. Some authors identified lower levels of 25-hydroxyvitamin D in patients with LP/OLP [140,141]. On the hand, vitamin D replacement in cases with ATDs might improve the outcome, but it does not overcome the standard approach [142,143].”

Added references:

Lebiedziński F, Lisowska KA. Impact of Vitamin D on Immunopathology of Hashimoto's Thyroiditis: From Theory to Practice. Nutrients. 2023;15(14):3174. doi:10.3390/nu15143174.

Carsote M, Paduraru DN, Nica AE, Valea A. Parathyroidectomy: is vitamin D a player for a good outcome? J Med Life. 2016;9(4):348-352.

Khozam SA, Sumaili AM, Alflan MA, Shawabkeh RAS. Association Between Vitamin D Deficiency and Autoimmune Thyroid Disorder: A Systematic Review. Cureus. 2022;14(6):e25869. doi:10.7759/cureus.25869.

Nosratzehi T. Serum vitamin D and antinuclear antibody level in oral lichen planus patients: a cross-sectional study. Ann Med Surg (Lond). 2023;85(2):136-139. doi:10.1097/MS9.0000000000000115.

Mahmoud SB, Anwar MK, Shaker OG, El Sharkawy DA. Possible Relation between Vitamin D and Interleukin-17 in the Pathogenesis of Lichen Planus. Dermatology. 2021;237(6):896-901. doi:10.1159/000510539. 

Bhakat B, Pal J, Das S, Charaborty SK, SircarMedical NR, Kolkata, RGKar, NorthBengal, Siliguri. A Prospective Study to Evaluate the Possible Role of Cholecalciferol Supplementation on Autoimmunity in Hashimoto's Thyroiditis. J Assoc Physicians India. 2023;71(1):1.

Dipasquale V, Lo Presti G, Milani GP, Corsello A, Agostoni C, Romano C. Vitamin D in Prevention of Autoimmune Diseases. Front Biosci (Landmark Ed). 2022;27(10):288. doi:10.31083/j.fbl2710288.

Section 4.2.3.D. Infectious diseases

….”coronavirus infection might trigger thyroid eye disease in GD or exacerbate TH in previously diagnosed (endocrine) patients since the virus might interfere with the immune and autoimmune systems, as well as the hormonal panel [159-161].”

Added references:

Brancatella A, Viola N, Santini F, Latrofa F. COVID-induced thyroid autoimmunity. Best Pract Res Clin Endocrinol Metab. 2023;37(2):101742. doi: 10.1016/j.beem.2023.101742.

Gorini F, Vassalle C. A Literature Review on SARS-CoV-2 and Other Viruses in Thyroid Disorders: Environmental Triggers or No-Guilty Bystanders? Int J Environ Res Public Health. 2023;20(3):2389. doi: 10.3390/ijerph20032389.

Lui DTW, Lee KK, Lee CH, Lee ACH, Hung IFN, Tan KCB. Development of Graves' Disease After SARS-CoV-2 mRNA Vaccination: A Case Report and Literature Review. Front Public Health. 2021;9:778964. doi: 10.3389/fpubh.2021.778964.

Thank you

…….

  • Environmental Elements (Lines 483-489): The section provides a comprehensive overview of environmental factors. However, separating information on smoking, vitamin D, infectious diseases, and nutrients into distinct subsections might enhance clarity.

Thank you very much. We already addressed this issue (kindly see above). Thank you

…….

  • (Non-Thyroid) Hormonal and Metabolic Potential Contributors (Lines 542-570): The discussion on vitamin D, IL-2, IL-4, IL-6, IL-10, and TNF-α cytokines is insightful. However, connecting these factors explicitly to both LP/OLP and thyroid conditions would strengthen the argument.

Thank you very much. We selectively highlight the connected data as shown belowe. We respectfully mention the fact that it does not represent the actual topic of the current work. Thank you

“Furthermore, vitamin D deficiency may potentially contribute to the development and OLP progression as it has a pivotal function in the regulation of inflammation and cytokine synthesis”

“Potential positive effects of its supplementation in OLP patients have been found”..

“vitamin D-associated immunosuppressive role has been acknowledged in HT: decreasing pro-inflammatory cytokine production, therefore inhibiting the immunological response mediated by cytokines, and suppressing the expression of HLA class II genes in the thyroid, leading to the inhibition of lymphocyte proliferation and inflammatory cytokines release..”

“IL-derivate panel is connected to LP/OLP and ATDs, for example, IL-8 in erosive OLP”

Thank you

…….

**5. Conclusions:** The conclusion summarizes the key findings. However, it would be beneficial to restate the main implications for clinical practice and potential avenues for future research.

Thank you very much. We adjusted the conclusion according to your recommendation, as already mentioned.

Moreover, we added: “Future research is mandatory to identify the underlying common pathogenic pathways and to highlight the connections between the thyroid anomalies and LP/OLP, the real epidemiologic impact of these connections, as well as the practical recommendations to be addressed amid daily practice. Whether a subgroup of individuals actually is affected by these dermatologic and endocrine diseases is yet to be detected.

Thank you

…….

Overall, the manuscript presents a thorough review of the association between LP/OLP and thyroid anomalies. Addressing the specific comments and recommendations for improvement should enhance the clarity and impact of the manuscript.

Thank you very much. We addressed each of your points. Thank you

…….

Comments on the Quality of English Language: Minor refinements required. Please adhere to clarity of terms and ensure consistency in terminology throughout the manuscript. Check for typographical errors and formatting inconsistencies.

Thank you very much. We re-edited the paper. Thank you

…….

Thank you very much!

Reviewer 2 Report

Comments and Suggestions for Authors

With pleasure, I read the paper titled “Crossroads between endocrine glands and skin: interplay of lichen planus with thyroid anomalies”. The topic is clinically relevant to practice, and of importance to the readers of “Biomedicines” journal.. Overall, the manuscript reads well and has good flow of ideas, up-to-date citations, and proper summary of data using tables. The authors investigated the relationship thyroid dysfunction with lichen planus. The manuscript is very detailed and provides good educational contents to the readers. The manuscript needs to be divided correctly into introduction, methods, results, discussion, and conclusion sections. A major issue is concerned about the methods used in the study and whether the study design is a systematic review or narrative review. The introduction section was detailed enough and provided adequate background to enlighten the readers about the purpose of the study. The methods section needs additional details for better completion. The results section needs some adjustment for complete reporting and clarification of some outcomes. However, the primary endpoints were reported adequately. The discussion section provided apt comparison with the published literature; however, it lacked highlighting the clinical implications, pinpointing the future directions, and acknowledging the limitations. The conclusion is line with the presented results; however, some edits may be needed. All in all, this manuscript is clinically significant and is very likely to be cited extensively in the future. I have some comments:

1.       TITLE. I recommend changing to “Crossroads between skin and endocrine glands: interplay of lichen planus with thyroid anomalies”. I just changed the sequence of the first part of the title to match that of the second part.

2.       ABSTRACT. It is very condensed, and it would be better if it can be shortened. All abbreviations upon first encounter should be spelled out fully. All numbers from 1 to 9 should be spelled out too. Please include a rationale of the study at the beginning, as well as include some proper conclusions at the end.

3.       INTRODUCTION. The authors need to (i) highlight the gap in literature, (ii) emphasize the significance of their study and whether it has been explored before or not, and (iii) conclude the section with some hypotheses.

4.       METHODS. The methods section needs to be clearly elaborated on and should be a stand-alone section. To me, this looks like a systematic review, and accordingly the authors need to follow the standard guidelines like PRISMA, MOOSE, or others. Why did only the authors search one database, PubMed? It would be better to screen other equally important databases, such as Scopus, Web of Science, Embase, and Google Scholar. This will ensure no key studies are missed. Were specific filters applied during database search based on year, language, or country of publication? The authors needs to present a PRISMA flow diagram for their study identification, study selection, and number of studies included in final review. The eligibility (inclusion and exclusion) criteria should be clearly specified. Similarly, the outcomes of the study should be clearly defined. Did you evaluate the quality of each study included in your narrative/systematic review? Did the authors attempt to quantify the results through a mean of meta-analysis, if applicable?

5.       RESULTS. The findings are very detailed, which is a big bonus. Table 1 should include additional important information such as country of publication, and other key demographics of patients (like gender, age, BMI, autoimmune disease, etc.), whenever applicable. Also, the use of letters like N N1, N2, is somehow confusing, but it should be OK. It would be recommended to summarize these findings too in Figures to add some sort of beautification and easy-to-grab summaries out of your article.

6.       DISUCSSION. The discussion section is extensive for those readers interested in in-depth knowledge. You need to start the section by providing a summary of the key findings. The strength of the article is ought to be mentioned. Additional limitations should also be highlighted such as lack to evaluate quality of included studies. Please complement the section with some discourses on clinical implications and future directions.

7.       CONCLUSION. Please summarize the 3-4 key findings first.

8.       OVERALL. The manuscript requires polishing for English language.

Comments on the Quality of English Language

Minor to moderate English editing is required.

Author Response

Response to Reviewer’ Comments - Reviewer number 2

Dear Reviewer,

Thank you very much for your time and your effort to review our manuscript.

We are very grateful for providing your valuable feedback on the article.

Here is our response and related amendment that has been made in the manuscript according to your review (marked in yellow colour).

With pleasure, I read the paper titled “Crossroads between endocrine glands and skin: interplay of lichen planus with thyroid anomalies”. The topic is clinically relevant to practice, and of importance to the readers of “Biomedicines” journal. Overall, the manuscript reads well and has good flow of ideas, up-to-date citations, and proper summary of data using tables. The authors investigated the relationship thyroid dysfunction with lichen planus. The manuscript is very detailed and provides good educational contents to the readers.

Thank you very much. We really appreciate it!

…….

The manuscript needs to be divided correctly into introduction, methods, results, discussion, and conclusion sections.

Thank you very much. We followed your recommendation and introduced the required sections. Thank you.

…….

A major issue is concerned about the methods used in the study and whether the study design is a systematic review or narrative review.

Thank you very much. This is a narrative review, as mentioned. To our aware, this is a standard type of scientific approach. We choose to introduce the data as a narrative review since various levels of statistical evidence are identified in the mentioned/cited papers. On the other hand, a systematic review pinpoints a specific critical assessment which in the matter of LP/OLP and thyroid conditions is rather limited so far.

However, this type of review is a well-recognized, traditional approach which is suitable for topics with less generous publications such as the update of the most recent data on LP/OLP and potential thyroid concerns. This allowed us to examine and evaluate the scientific panel on this specific cross-disciplinary topic in a useful way for various practitioners with different backgrounds including dermatology and endocrinology. 

The study design has been specified at Methods (section 2).

This aspect has also been displayed at the Discussion (section 4) as follows:

“As limitations of the current work we mention the design in terms of narrative review...”

Thank you

…….

The introduction section was detailed enough and provided adequate background to enlighten the readers about the purpose of the study.

Thank you very much.

…….

The methods section needs additional details for better completion.

Thank you very much. We extended this section with additional information, as following:

“The inclusion criteria were: original, clinical (non-experimental) studies in humans (clinical studies); English papers; studies that provided the traditional thyroid panel of evaluation (any of the followings: thyroid function, serum antibodies, imaging features such as ultrasound, and pathological report in some cases). Exclusion criteria were: animal studies, interventional trials; non-English papers, and other types of articles such as case reports, editorials, letters to editor, reviews.

We followed three main sections: LP/OLP and anomalies of the thyroid in terms of dysfunctionality (hypothyroidism and hyperthyroidism), thyroid autoimmunity (ATDs, namely HT and GD) and neoplasia (nodules and/or cancer). Particularly, the data were displayed according to the followings subsections: the association between LP/OLP and thyroid function anomalies (6 confirmatory studies and 5 non-confirmatory cohorts); the spectrum of positive thyroid autoimmunity amid LP/OLP (8 confirmatory and one non-confirmatory study); the use of levothyroxine replacement for prior hypothyroidism in patients confirmed with LP/OLP (5 studies), and the co-presence of thyroid nodules/cancer concerning the same dermatologic conditions (2 studies).”

Thank you

…….

The results section needs some adjustment for complete reporting and clarification of some outcomes. However, the primary endpoints were reported adequately.

Thank you very much. We revisited the Results section. Thank you

…….

The discussion section provided apt comparison with the published literature; however, it lacked highlighting the clinical implications, pinpointing the future directions, and acknowledging the limitations.

Thank you very much. We expanded these data at Discussion section as following:

Clinical implications: “Potential clinical implications of the current data with respect to multiple connections between thyroid conditions of various types and the presence of LP and/or OLP involve firstly awareness that the same population group might be affected by the dermatologic and endocrine diseases (simultaneously or not, across life span); secondarily, some patients with LP/OLP might require a detailed hormonal evaluation and consecutive follow-up if thyroid anomalies are identified as, for example, the need of lifelong levothyroxine requirement or the increased risk of developing hypothyroidism in individuals with positive blocking antibodies; and, finally, we should consider that autoimmune thyroiditis and LP/OLP might belong to the same cluster of general autoimmune diseases and other organs might be targeted, as well, under these circumstances.“

Limitations: “As limitations of the current work we mention the design in terms of narrative review according to a single database research (without a specific evaluation of the quality of each included study) which, however, allowed us a flexible approach of various sub-chapters underlying different levels of statistical evidence amidst analysing the LP/OLP and thyroid connections. As mentioned, some important topics such as thyroid dysfunction and positive autoimmunity in patients diagnosed with LP/OLP are linked to both confirmatory and non-confirmatory studies; that is why a clear conclusion still represents an open issue at this point. Moreover, many data actually involved the individuals with OLP, rather than LP, thus pinpointing the fact that two distinct subgroups of individuals should be taken into consideration when analysing the thyroid profile, a specific issue which is yet to be determined.”

Future expansion of this research: ”Multiple clinical data are not convincing yet, as seen with regard to the thyroid nodules and malignancies in OLP/LP. Further pathogenic studies will bring an expansion to the current knowledge and understanding of common traits as discussed above. Awareness of potential thyroid damage is necessary in LP/OLP and interventional (longitudinal) studies are mandatory to display the long term outcome in patients diagnosed with both conditions and potential interferences of specific therapies. Moreover, inter-disciplinary algorithms involving the panel of baseline assessments and long term follow-up of subjects suffering from LP/OLP and thyroid anomalies should help the clinicians with different backgrounds.

Thank you

…….

The conclusion is line with the presented results; however, some edits may be needed.

Thank you. We extended the data within the Conclusions section. Thank you.

…….

All in all, this manuscript is clinically significant and is very likely to be cited extensively in the future.

Thank you very much.

…….

I have some comments: TITLE. I recommend changing to “Crossroads between skin and endocrine glands: interplay of lichen planus with thyroid anomalies”. I just changed the sequence of the first part of the title to match that of the second part.

Thank you very much. This is a very useful recommendation. Thank you

…….

ABSTRACT. It is very condensed, and it would be better if it can be shortened. All abbreviations upon first encounter should be spelled out fully. All numbers from 1 to 9 should be spelled out too. Please include a rationale of the study at the beginning, as well as include some proper conclusions at the end.

Thank you very much. We corrected the abstract. Thank you

…….

INTRODUCTION. The authors need to (i) highlight the gap in literature, (ii) emphasize the significance of their study and whether it has been explored before or not, and (iii) conclude the section with some hypotheses.

Thank you very much. We expanded these data at Introduction. For instance, we added: ”The motivation of this topic is based on the current gap in understanding the connection between LP/OLP and thyroid panel. According to nowadays knowledge, LP/OLP does not represent a single disease with a clear pathogenic component, but rather a type of immune response to a multitude of environmental elements (which are more or less understood so far) in genetically susceptible subjects. Whether thyroid anomalies are part of these LP contributors or they are also the consequence of the same environmental factors is less clear. A part from the pathogenic loops, our hypothesis involves clinical aspects, namely the fact that some population subgroups diagnosed with LP are at higher risk of having hypothyroidism, as well as positive thyroid serum antibodies than controls.”

Thank you

…….

METHODS. The methods section needs to be clearly elaborated on and should be a stand-alone section. To me, this looks like a systematic review, and accordingly the authors need to follow the standard guidelines like PRISMA, MOOSE, or others. Why did only the authors search one database, PubMed? It would be better to screen other equally important databases, such as Scopus, Web of Science, Embase, and Google Scholar. This will ensure no key studies are missed. Were specific filters applied during database search based on year, language, or country of publication? The authors needs to present a PRISMA flow diagram for their study identification, study selection, and number of studies included in final review. The eligibility (inclusion and exclusion) criteria should be clearly specified. Similarly, the outcomes of the study should be clearly defined. Did you evaluate the quality of each study included in your narrative/systematic review? Did the authors attempt to quantify the results through a mean of meta-analysis, if applicable?

Thank you very much. The type of the study design and the improvement of the Methods section, including inclusion/exclusion criteria have been discussed above. Since this is a narrative review, a flexible research is allowed, that is why we used only PubMed. However, the Introduction and Discussion sections are not limited to a single database as pointed out by the entire panel of references that are 177. The timeframe of research, the language, the key research terms have already been provided. PRISMA stands for “Preferred Reporting Items for Systematic Reviews and Meta-Analyses” (http://prisma-statement.org/prismastatement/flowdiagram.aspx) and this is a narrative review, not a systematic one. We already explained the fact the collected data were not homogenous, thus a systematic review, neither a meta-analysis was intended since we intended a large panel of thyroid approach. At limitations within Discussion section, the fact that the quality of the collected studies has not been assessed was mentioned, too. Thank you

…….

RESULTS. The findings are very detailed, which is a big bonus. Table 1 should include additional important information such as country of publication, and other key demographics of patients (like gender, age, BMI, autoimmune disease, etc.), whenever applicable.

Thank you very much. We added the requested information (if available). Notably, body mass index does not represent a variable that has been specifically addressed amid these studies despite being a very interesting observation. Thank you

For instance, we included such parameters as the followings:

  1. LP/OLP and thyroid dysfunction: confirmatory studies

Piloni

2023

[43]

Retrospective, case-control study (#)

Italy

N=307 patients

158 (51.47%) F + 149 (48.53% M)

(age>18 y; range: 49-75 y, mean age of 63 y)

N1=76 OLP patients

N2=133 OLL patients

N3=98 controls

Amato-Cuartas

2019

[44]

Cross-sectional study

Colombia

N=860 patients

N1=14 OLP patients

(representing 1.6% of N, mean age of 59.6 y, F= 11/16)

Kumar

2019

[45]

Case-control study

India

N=75 LP patients (adults with at least 3-month disease duration)

43 (57.33%) F and 32 (42.66%) M

age: 16-45 y (mean age of 35.2 y)

17.3% with metabolic syndrome

N1=12 OLP patients

N2=75 controls

Robledo-Sierra

2018

[46]

Case-control study

Sweden

N=110 OLP+/LT4-

99 (90%) F

(mean age of 65.6 y)

N1=657 healthy controls

Robledo-Sierra  2015

[47]

Case-control study

Sweden

(#,#)

N=1611 OLP patients

(data collection from general population: standardized registration method)

N1=1615 controls

Siponen

2010

[48]

Retrospective case-control study

Finland

(#,#,#)

N=222 OLP/OLL patients

N1=222 controls

  1. LP/OLP and thyroid dysfunction: non-confirmatory studies

Tang

2020

[49]

Cross-sectional study

China

N=585 OLP patients

167 (87.97%) F + 23 (12.1% M)

(age>18 y; mean age of 52.85 ± 13.15 y)

N1=10.441 controls

Kats

2019

[50]

Retrospective, case-control study Israel

(#,#,#,#)

N=102 OLP patients

70.6% F

(mean age of 55.7 ± 13.2 y)

N1=102 controls

Zhou

2018

[51]

Case-control study

China

(#,#,#,#,#)

N=192 OLP patients

75.5% F + 24.45% M

(mean age of 49.53 ± 9.93 y)

N1=123 OLL patients

N2=163 controls

Lavaee   

2016

[52]

Retrospective, comparative study

Iran

N=523 OLP patients

387(74%) F + 136 (26.2%) M

(age>18 y; mean age of 48.43 ± 13.82 y)

N1=523 controls

  1. LP/OLP and positive thyroid autoimmunity: confirmatory studies

Anttonen

2023

[53]

Retrospective study

Northern Finland

N=619 LP patients

58.3% F

(mean age of 54.2 y)

N1=258 LP males (41.7%)

N2=361 LP females (58.3%)

Zhang  

2022

[56]

Cross-sectional study

China

N=247 OLP patients

186 (75.3%) F + 61 (24.7%) M

(mean age of 45.21 ± 12.72  y)

N1=61 OLP males (24.70%)

N2=186 OLP females (75.30%)

Alikhani

2017

[59]

Case-control study

Iran

N= 92 OLP patients

34 (74%) F

(mean age of 45 y, ranges: 17-68)

N1=44 EOLP patients

N2=48 non-EOLP patients

Lo Muzio

2013

[64]

Cross-sectional study

Italy

N=105 OLP patients

N’=15/105 patients with OLP+HT (14/15 F +1/15 M; age ranges:33-63 y)

Of note, the autoimmune panel is already introduced within the fourth column, as following:

LP/OLP and thyroid dysfunction: confirmatory studies

Piloni

2023

[43]

Retrospective, case-control study (#)

N=307 patients

N1=76 OLP patients

N2=133 OLL patients

N3=98 controls

Patients with HT:

N1’=15/76 patients (19.74%)

N2’=15/133 patients (11.28%)

N3’=5/98 patients (5.1%)

N1’ vs. N3’: p=0.005

N2’ vs. N3’: p=0.1075

Anttonen

2023

[53]

Retrospective study

N=619 LP patients N1=258 LP males (41.7%)

N2=361 LP females (58.3%)

Patients with ATD:

N’=12.44% patients

N1’=5% patients

N2’=17.7% patients

N1’ vs. N2’: p<0.001

Zhang  

2022

[56]

Cross-sectional study

N=247 OLP patients

N1=61 OLP males (24.70%)

N2=186 OLP females (75.30%)

Patients with HT:

N’=98/247 patients (39.67%)

N1’=12/61 patients (19.67%)

N2’=86/186 patients (46.24%)

N1’ vs. N2’: p=0.000

Tang

2020

[57]

Cross-sectional study

(#,#,#,#,#,#)

N=585 OLP patients

N1=10,441 controls

Patients with HT:

N’=71/585 patients (12.1%)

N1’=638/10.441 patients (6.1%)

N’ vs. N1’: p<0.05

Zhou

2018

[58]

Case-control study

N=192 OLP patients

N1=123 OLL patients

N2=163 controls

Patients with HT:

N’=40/192 patients (20.83%)

N1’=23/123 patients (18.69%)

N’ vs. N1’: p=0.011

N2’=16/163 patients (9.81%)

N’ vs. N2’: p=0.000

Alikhani

2017

[59]

Case-control study

N= 92 OLP patients

N1=44 EOLP patients

N2=48 non-EOLP patients

Patients with TPOAb +ve status:

N1’=30/44 patients (68%)

N2’=33/48 patients (33%)

N1’ vs. N2’: p=0.023

Robledo-Sierra

2015

[47]

Case-control study (#,#)

N=108 OLP patients

N1=40 controls

Patients with HT:

N’=19/108 patients (22.1%)

N1’=1/40 patients (2.5%)

N’ vs. N1’: p=0.0037

Lo Muzio

2013

[64]

Cross-sectional study

N=105 OLP patients

Patients with HT:

N’=15/105 patients (14.3%)

N’ vs. general population: p<0.0003 (general population with a HT-related hypothyroidism prevalence of 1%)

LP/OLP and positive thyroid autoimmunity: non-confirmatory studies

Robledo-Sierra 2018

[46]

Case-control study

N=110 OLP+/LT4- N1=657 controls

N2=108 OLP+/LT4+

N3=58 OLP-/LT4+

Increased levels of TPOAb:

N’=9/110 patients (9.4%)

N1’=99/657 patients (15%)

N’ vs. N1’: p=0.024

Thank you

…….

Also, the use of letters like N N1, N2, is somehow confusing, but it should be OK.

Thank you very much. “N” stands for the number of patients and it has been introduced under Table 1 and within the Table of Abbreviations. Thank you

…….

It would be recommended to summarize these findings too in Figures to add some sort of beautification and easy-to-grab summaries out of your article.

Thank you very much for this interesting suggestion. We added the key findings in the Figure below which was displayed at the end of Discussion section after all the cited articles has been displayed and discussed. Thank you.

Figure 1. Key findings across mentioned studies with respect to LP/OLP and thyroid connections [43-53,56,64,71,72]

…….

DISUCSSION. The discussion section is extensive for those readers interested in in-depth knowledge. You need to start the section by providing a summary of the key findings.

Thank you very much. The main findings are introduced in section 4.1. Thank you

…….

The strength of the article is ought to be mentioned.

Thank you very much. We added the followings: “Overall, the strength of the current work is represented by the complexity of these data coming from both dermatologic and endocrine domains; this sample – based analysis being one of the most complexes of this type, to our aware. Modern medical era, while being focused on an individual approach, will bring a more specific frame regarding the thyroid profile in LP/OLP or, quite the other way, regarding the LP/OLP subgroup among subjects diagnosed with various thyroid anomalies.”

Thank you

…….

Additional limitations should also be highlighted such as lack to evaluate quality of included studies.

Thank you very much. We included this aspect, too. Thank you

…….

Please complement the section with some discourses on clinical implications and future directions.

Thank you very much. These aspects have already been discussed above. Thank you

…….

CONCLUSION. Please summarize the 3-4 key findings first.

Thank you very much. We adjusted the Conclusion. We respectfully need to take into consideration the recommendations of the reviewer number, as well. Thank you

…….

OVERALL. The manuscript requires polishing for English language.

Thank you very much. We re-edited the paper. Thank you.

…….

Thank you very much.

Reviewer 3 Report

Comments and Suggestions for Authors

The authors reviewed studies on the link of lichen planus and thyroid disorders or thyroid autoimmunity. This is a difficult topic because the etiology of lichen planus is not clear. Studies are controversial and the authors did their best to find explanations for that.

The reader would have liked to receive a clear take home message but given the available studies, this appears unrealistic. Nevertheless, this lowers the significance of the contribution to the field.

Despite of that, the review is a good summary of existing studies.

Why did the authors perform a narrative review instead of a systematic review or a meta-analysis with clear inclusion and exclusion criteria, quality assessment and analysis of homogeneity?

Few errors should be corrected, e.g. in l.245-246, l.317, l.553

Author Response

Response to Reviewer’ Comments – Reviewer number 3

Dear Reviewer,

Thank you very much for your time and your effort to review our manuscript.

We are very grateful for providing your valuable feedback on the article.

Here is our response and related amendment that has been made in the manuscript according to your review (marked in yellow color).

The authors reviewed studies on the link of lichen planus and thyroid disorders or thyroid autoimmunity. This is a difficult topic because the etiology of lichen planus is not clear. Studies are controversial and the authors did their best to find explanations for that.

Thank you very much.

…….

The reader would have liked to receive a clear take home message but given the available studies, this appears unrealistic. Nevertheless, this lowers the significance of the contribution to the field. Despite of that, the review is a good summary of existing studies.

Thank you very much. We introduced the key findings at the begging of each subsection within the Results section, as well as subsection 4.1. from Discussion. Thank you

…….

Why did the authors perform a narrative review instead of a systematic review or a meta-analysis with clear inclusion and exclusion criteria, quality assessment and analysis of homogeneity?

Thank you very much. This is a narrative review, indeed, as mentioned. To our aware, this is a standard type of scientific approach. We choose to introduce the data as a narrative review since various levels of statistical evidence are identified in the mentioned/cited papers. On the other hand, a systematic review/meta-analysis pinpoints a specific critical assessment which in the matter of LP/OLP and thyroid conditions is rather limited so far. However, this type of review is a well-recognized, traditional approach which is suitable for topics with less generous publications such as the update of the most recent data on LP/OLP and potential thyroid concerns. This allowed us to examine and evaluate the scientific panel on this specific cross-disciplinary topic in a useful way for various practitioners with different backgrounds including dermatology and endocrinology. 

The study design has been specified at Methods (section 2).

This aspect has also been displayed at the Discussion (section 4) as follows:

“As limitations of the current work we mention the design in terms of narrative review...”

The inclusion/exclusion criteria have been provided as followings:

“The inclusion criteria were: original, clinical (non-experimental) studies in humans (clinical studies); English papers; studies that provided the traditional thyroid panel of evaluation (any of the followings: thyroid function, serum antibodies, imaging features such as ultrasound, and pathological report in some cases) that were published within the mentioned time frame. Exclusion criteria were: animal studies, interventional trials; non-English papers, and other types of articles such as case reports, editorials, letters to editor, reviews.

At limitations within Discussion section, the fact that the quality of the collected studies has not been assessed was mentioned, too.

Thank you

…….

Few errors should be corrected, e.g. in l.245-246, l.317, l.553

Thank you very much. We corrected them. Thank you

…….

Thank you very much

Round 2

Reviewer 2 Report

Comments and Suggestions for Authors

With pleasure, I enjoyed reading the revised the draft of the manuscript. The authors did a great job by adequately addressing all the comments. The manuscript now reads very well and is scientifically valid, methodologically solid, and intellectually curious. It hold important clinical implications. All in all, I recommend the manuscript for acceptance in its present form, subject to routine English copyediting by the editorial team.

Comments on the Quality of English Language

Minor English editing may be required during copyediting